Small scale spatial variability of bare-ice reflectance at Jamtalferner, Austria

Lea Hartl (1), Lucia Felbauer (1), Gabriele Schwaizer (2), Andrea Fischer (1)

1) Institute for Interdisciplinary Mountain Reasearch, Austrian Academy of Sciences, Technikerstraße, 21a, ICT, 6020 Innsbruck, Austria
2) ENVEO GmbH, Fürstenweg 176, 6020 Innsbruck, Austria

**Abstract**

As Alpine glaciers become snow free in summer, more dark, bare ice is exposed, decreasing local albedo and increasing surface melting. To include this feedback mechanism in models of future deglaciation, it is important to understand the processes governing broadband and spectral albedo at a local scale. However, little in situ reflectance data has been measured in the ablation zones of mountain glaciers. As a contribution to this knowledge gap, we present spectral reflectance data (Hemispherical-Conical-Reflectance-Factor) from 325 to 1075 nm collected along several profile lines in the ablation zone of Jamtalferner, Austria. Measurements were timed to closely coincide with a Sentinel-2 and Landsat-8 overpass and are compared to the respective ground reflectance (Bottom-Of-Atmosphere) products. The brightest spectra have a maximum reflectance of up to 0.7 and consist of clean, dry ice. In contrast, reflectance does not exceed 0.2 for dark spectra where liquid water and/or fine-grained debris are present. Spectra can roughly be grouped into dry ice, wet ice, and dirt/rocks, although gradations between these groups occur. Neither satellite captures the full range of in situ reflectance values. The difference between ground and satellite data is not uniform across satellite bands, between Landsat and Sentinel, and to some extent between ice surface types (underestimation of reflectance for bright surfaces, overestimation for dark surfaces). We highlight the need for further, systematic measurements of in situ spectral reflectance properties, their variability in time and space, and in-depth analysis of time-synchronous satellite data.

# 1. Introduction
## 1.1 General context and aims
Under ongoing climate change, mountain glaciers are retreating at unprecedented rates (Zemp et al, 2015, 2019). Glaciers in the Eastern Alps are losing mass rapidly, and due to persistent loss of snow cover exposing the underlying firn (Fischer, 2011), many have lost much of their firn cover. An increasing amount of darker bare ice is exposed in Summer and at some glacier tongues, darkening of the ice has been observed (Klok et al., 2003). These feedback mechanisms in turn increase the amount of energy absorbed and accelerate melt (e.g. Paul et al., 2005; Box et al., 2012; Naegeli et al., 2017 & 2019). The reflective properties of glacier ice are affected by e.g. the absence or presence and amount of dust, pollen, debris, cryoconite, supraglacial water, and biota including local production rates (Dumont et al., 2009; Gabbi et al. 2015; Azzoni et al., 2016). Variability is understood to be high, but few measurements and models exist. In a glaciological context, the spatial and temporal variability of ice albedo is understudied compared to snow albedo.

We present spectroradiometric data on the spatial variability of bare-ice reflectance at the tongue of Jamtalferner, Austria, aiming to contribute to closing the knowledge gap in bare ice variability as an important feedback mechanism in glacier mass loss. Specifically, we aim to:

1) Provide a first-order quantitative assessment of spatial variability of surface reflectance in the ablation area of the rapidly melting Jamtalferner, quantifying possible ranges of spectral reflectance and qualitatively summarizing different surface types.
2) Compare commonly used reflectance products derived from Landsat-8 and Sentinel-2 data with in situ measurements, highlighting areas in which further study is required if ongoing processes related to deglaciation are to be fully captured by satellite data.

## 1.2 In situ and remote sensing-based change detection of surface reflectance properties of glacier ice

In the following section we summarize previous studies on this topic. For clarity, we begin with a note on terminology: Following the definitions and guidelines detailed in Schaepman-Strub et al., (2004, 2006) and Nicodemus et al. (1977), we use the term "albedo" for bihemispherical reflectance (BHR), including cases where this parameter is approximately measured with an albedometer. In situ measurements with field spectrometers – such as they were carried out for this study – generally represent Hemispherical-Conical-Reflectance-Factors (HCRF). For exact specifications of what is represented by satellite derived surface reflectance products we refer to the documentation of the respective products as this differs between sensors and product suites.

While it is generally understood that albedo is a major driving factor for the energy balance and radiative regime of
glaciers, few studies discuss ice albedo and its variability at the local level. Early investigations of ice albedo were
carried out by Sauberer in 1938. Building on this work, Sauberer and Dirmhirn (1951) showed that albedo is highly
variable in time and space and strongly affects the radiation balance. They reported mean values of 0.37 for clean ice
and 0.13 for dirty ice at Sonnblick glacier (Austria), a pronounced diurnal cycle of albedo related to refreezing of the
surface, and influence of wind transported fine mineral dust. In another study based on measurements at Sonnblick, they
highlighted that the collection of mineral dust in cryoconite holes affects albedo, as does liquid water, and showed a
diurnal reduction of albedo of about 0.2 under clear sky conditions, which they attribute to melt-freeze cycles on the ice
surface (Sauberer and Dirmhirn, 1952). Jaffé (1960) also pointed out the importance of cryoconite and air content in the
upper most ice layer for the radiative properties. Dirmhirn and Trojer (1955) presented a histogram-like curve of the
frequency of different ice albedo values measured on the tongue of Hintereisferner (Austria): Broadband ice albedo
ranges from <0.1 to about 0.58, with a frequency maximum at 0.28. Similar to the results from Sonnblick, melt-related
diurnal albedo variations were also found at Hintereisferner. In a detailed study of the radiation balance at
Hintereisferner, Hoinkes and Wendler (1968) showed the importance of summer snow falls for albedo, as well as
seasonal changes in ice albedo, and their significant contribution to ablation.
Considering the growing dominance of bare ice areas both compared to overall glacier area and in terms of glacier-wide
mass- and energy balance, the sensitivity of the latter parameters to changing reflectance properties has become of
increasing interest throughout approximately the last decade. Using a combination of mass balance data from multiple
Swiss glaciers and the Landsat-8 surface reflectance product, Naegeli and Huss (2015) show that mass balance
decreases on average by 0.14 m w.e. a−1 per 0.1 albedo decrease. In order to better delineate associated driving
processes at the glacier surface, it is important to assess reflectance properties not only as broadband albedo at the scale
of a glacier, but at a high spectral and spatial resolution. A number of studies attribute recent darkening of European
glaciers to increased accumulation of mineral dust (e.g. Oerlemans et al., 2009, Azzoni et al., 2016) and black carbon
(e.g. Painter et al., 2013, Gabbi et al., 2015). Similar findings have been reported from the Himalayas (e.g. Ming et al.
2012, 2015; Qu et al. 2014) and the Greenland ice sheet (Dumont et al., 2009). Some discussion remains as to whether
the observed darkening is primarily due to the increase of bare ice areas compared to overall glacier area, or whether
there is a darkening of the bare ice areas as such, and if so, whether bare ice areas are darkening due to local processes
or large scale systemic change (e.g. Box et al., 2012; Alexander et al., 2014; Naegeli, 2019).
Different methodological approaches have been used to address specific changes in the surface characteristics of the
ablation zone as they relate to changes in reflectance properties and energy absorption across the electromagnetic
spectrum: Using both hyperspectral satellite data and in situ HCRF measurements, Di Mauro et al. (2017) find that the
presence of elemental and organic carbon leads to darkening of the ablation zone at Vadret da Morteratsch (Switzerland)
and discuss potential anthropogenic contributions. Azzoni et al. (2016) use semi-automatic image analysis techniques
on photos of the ice surface at Forni glacier (Italy) to quantify the amount of fine debris present on the surface and its
effect on the albedo. They find an overall darkening due to increasing dust, as well as significant effects of melt and rain
water. Naegeli et al. (2015) use in situ spectrometer and airborne image spectroscopy data with a pixel resolution of
approximately 2m to classify glacier surface types and map spectral albedo on Glacier de la Plaine Morte in
Switzerland. Additionally, they highlight the difference in scale between albedo variability at the ice surface and the
pixel resolution of satellite data and the need for detailed case studies combining ground truth data and remote sensing
techniques to bridge this gap. In situ data is also essential for model verification, as shown e.g. by Malinka et al. (2016),
who use reflectance spectra (HCRF) gathered on sea ice to validate modelled reflectance parameters.
In order to scale assessments of ice albedo from the local to a regional or global level, satellite-derived data are
indispensable. Earlier in the satellite era, several studies carried out comparisons of albedo data measured on the ground
and surface reflectance derived from Landsat-5 Thematic Mapper scenes, finding considerable differences between in
situ and satellite data especially in the ablation area (e.g. Hall et al., 1989 & 1990; Koelemeijer et al., 1993; Winther,
1993; Knap et al., 1999). These works are mostly based on albedo data from a single location, such as an automatic
weather station (AWS), and it was often not possible to carry out ground measurements so that they coincided with the
satellite overpasses. More recently, Brun et al. (2015) highlight the importance of remote sensing data for monitoring of
glacier albedo changes in remote regions where data collection on the ground is impossible or impractical and compare
MODIS data with in situ radiation measurements. Albedo measurements from AWS sites on the Greenland ice sheet –
associated with the PROMICE and GC-Net monitoring networks - have been used to improve gridded albedo products
based on MODIS data, showing the importance of using ground truth in conjunction with satellite data (Box et al.,
2013; van As et al., 2017). Narrow-to-broadband conversions remain a challenge in this regard and commonly used
conversions are typically designed for use with Landsat-5 or 7, rather than Landsat-8 or Sentinel-2, which increases the
uncertainties inherently associated with any narrow-to-broadband conversion (Gardner et al, 2010; Naegeli et al., 2017).
In addition, studies assessing the potential effects of anisotropy on satellite-derived surface reflectance data are sparse
and the magnitude of associated uncertainties is hard to quantify (Naegeli et al., 2015 & 2017).
Naegeli et al. (2019) quantify trends in bare ice albedo for 39 Swiss glaciers using Landsat surface reflectance data
products for a 17-year period. While they do not find a clear, wide spread darkening trend of bare ice surfaces
throughout the entirety of their data set, they note significant negative trends at the local level, most notably for certain
terminus areas. A detailed comparison of different albedo products derived from airborne imaging spectroscopy (APEX)
and Landsat and Sentinel data by Naegeli et al. (2017) further highlights the gap between albedo variability on the
ground and its representation in remote sensing data of varying resolution. A recent study by Di Mauro et al. (2020)
uses in situ HCFR data and DNA analysis to show that ice algae affect albedo on a Swiss glacier.
Despite the growing body of work on this topic (see Table 1), reflectance properties – spectral as well as broadband,
local as well as regional, short time as well as seasonal - remain understudied compared to other parameters routinely
recorded at Jamtalferner and other long-term glaciological monitoring sites. However, surface changes and associated
changes of the spectral characteristics in the ablation area (e.g. due to debris cover, supraglacial meltwater, deposition of
impurities) are expected to play a significant role in determining the future development of these glaciers. Incorporating
relevant parameters into monitoring efforts is highly desirable. The accuracy of direct measurements of mass balance
depends on the representation of all surface types in the stake network, and the correct attribution of unmeasured areas
to measured stake ablation. Accordingly, a better understanding of how surface types differ in terms of their reflective
properties is required to maintain the stake network on a rapidly changing glacier. To this end, it is important to
understand whether satellite-derived data can provide a basis for defining surface classes to be covered by stakes, or
whether it does not allow for the retrieval of the full bandwidth of reflectance variability relevant to the ice melt rate. In
addition, delineating the temporal variability of reflectance properties is relevant to degree day modelling, as a changing
albedo would alter parameters in the model.
Table 1: Measurements of bare ice reflectance properties on mountain glaciers: Overview.

| Glacier | Albedo type | Temporal resolution | Spatial resolution | Reference |
|---|---|---|---|---|
| Hintereisferner, AT | Total | Multiple days | Multiple points on different surface types | Dirmhirn and Trojer, 1955. |
| Hintereisferner, AT | Total | Multiple times on one day | 2 points | Jaffé, 1960. |
| Northern China (glacier not specified) | Spectral | Not specified | Different surfaces | Zeng et al., 1984. |
| Forbindels, Greenland | Spectral | One measurement campaign | Regular grid of points around multiple study sites | Hall et al., 1990. |
| Hintereisferner, AT | Spectral | 7 days during ablation season | Points along a profile | Van de Wal et al., 1992. |
| Austre Brøggerbreen, Midre Lovénbreen, Svalbard | Spectral, total shortwave | Multiple days during ablation season | 1 point | Winther, 1993. |
| Morteratsch, CH | Narrow band (Landsat TM bands 2 and 4) | One measurement campaign | Multiple points | Greuell and de Wildt, 1999. |
| Haut Glacier d'Arolla, CH | Total | One measurement campaign | Multiple points | Knap et al., 1999. |
| Hintereisferner, AT | Spectral | One measurement campaign | Multiple points | Hendriksa et al., 2003 |
| Morteratsch, CH | Total | Continuous AWS measurements | Multiple AWS locations | Klok et al., 2003 |
| Chhota Shigri, Mera Glaciers, Nepal | Total shortwave | Continuous AWS measurements | AWS location | Brun et al., 2015. |
| Forni Glacier, IT | Total | Multiple measurements during multiple years | Multiple points | Azzoni et al., 2016. |

| Glacier de la Plaine Morte, CH | Spectral | One measurement campaign | Multiple points | Naegeli et al., 2015. |
|---|---|---|---|---|
| Findelen, CH | Total | Continuous AWS measurements | AWS location | Naegeli et al., 2017. |
| Morteratsch, CH | Spectral | One measurement campaign | Multiple points | Di Mauro et al., 2017; Di Mauro et al, 2020 |
| Greenland ice sheet | Total | Continuous AWS measurements | Multiple AWS locations | van As et al., 2017; Box et al., 2013 |
| De Geerfonna and Elfenbeinbreen, Svalbard | Total | Continuous AWS measurements | 1 AWS on each glacier | Möller and Möller, 2017 |
| Jamtal, AT | Spectral | One measurement campaign | Multiple points | This study |


**2. Data, Methods, and Study Site**
**2.1. Study site – glaciological background**
Jamtalferner was chosen for this study as it has the smallest end-of-season snow cover amongst the glaciers with long
term mass balance monitoring in Austria. Jamtalferner is located in the Silvretta mountain range, which intersects the
border between Austria and Switzerland. Jamtalferner is the largest glacier on the Austrian side of Silvretta (Fig. 1, size
in 1970: 4.115km², size in 2015: 2.818km²). The history of scientific research at the site goes back as far as 1892, when
length change measurements were first carried out, and a wealth of cartographic, geodetic, and glaciological data are
available (Fischer et al., 2019). Orthophotos and cartographic analysis show that debris cover at the glacier terminus
and in the lower elevation zones has increased (debris covered percentage of total area: 1.7% in 1970, 24.1% in 2015),
while firn cover is decreasing (firn covered area in 1970: 75%, in 2015: 13%, mean accumulation area ratio (AAR)
1990/91-99/00: 0.35, mean AAR 2010-2017/18: 0.12, Fischer et al., 2016).
Mass balance measurements via the direct glaciological method began in 1988/1989. In recent years, increasing mass
loss was recorded across all elevation zones (Fischer et al., 2016). The lowest elevation zones are dominant in terms of
total ablation and thus net balance. Melt in the lowest altitudes has been increasing during the last two decades of
negative mass balances and the variability of surface albedo at and near the glacier terminus affects melt over the full
duration of the ablation season.

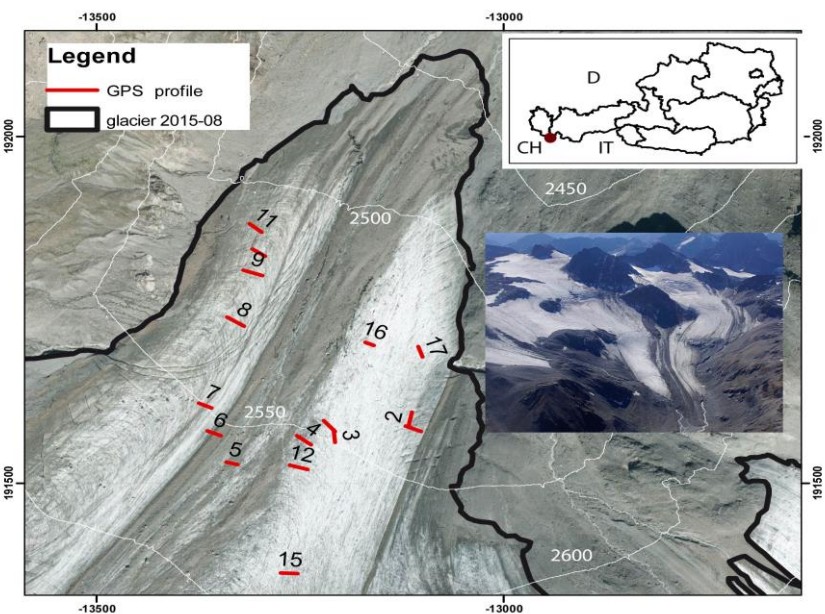

Figure 1: Tongue of Jamtalferner glacier (Orthophoto, August 2015, Source: Tyrolean Government/ TIRIS) with profile
lines of spectroadiometer measurements indicated in red. Insert: Aerial photograph of Jamtalferner, 20.09.2018 (Photo:
Andrea Fischer).
**2.2 In situ measurements of spectral reflectance**
The field campaign was carried out on September 4th, 2019. This date was selected for two reasons: Favourable weather
conditions and temporal proximity to overpasses of both Sentinel-2 (on the same day) and Landsat-8 (on September
3rd). With a large area of high pressure over western and central Europe, the weather at the study site was sunny and dry
on throughout Sept. 3rd and 4th. Using an ASD Field Spec Handheld 2 spectroradiometer (ASD Inc., 2010), a total of
246 reflectance spectra (HCFR) was collected, with 12 spectra measured at point locations and 234 spectra measured
along 16 profile lines. Profiles were measured along a 20m measuring tape in such a way that individual spectra were
gathered at equal intervals, with 14 profile lines containing 11 spectra spaced at 2 m. 2 profiles contain 40 spectra –
these were also gathered at equal intervals but with a higher resolution. Measurements began at 08:28 GMT (10:28 local
time) and ended at 13:43 GMT. The coordinates of the start and end points of each profile line, as well as any spectra
measured outside of the lines, were recorded with a Garmin etrex VISTA HCx, a standard handheld GPS device, which
also recorded the time of day. The horizontal accuracy of the GPS coordinates is better than 3 m as per the internal
accuracy assessment of the GPS device. The timestamps of the GPS points for the start and end points of the profiles
were used to compute solar elevation and azimuth. For each profile, the mean solar elevation and azimuth between the
respective start and end points is given in Table 2. Measurements were taken 35 cm above ground from nadir with a
bare fibre optic cable without additional fore-optics. Test measurements in the field showed high consistency between
multiple measurements at the same point, so that we chose to use single measurements at each location rather than
average over multiple measurements. The instrument was handheld and not mounted on a stand to minimize shading.
This measurement set up is similar to that of previous studies (Naegeli, 2015; Di Mauro et al., 2017) and yields a
circular field of view (FOV) with a radius of approximately 7.8 cm for flat ground. The instrument operates between
325 and 1075 nm with an accuracy of $\pm 1$ nm and a resolution of $<3$ nm at 700 nm. We used a feature of the instrument
that allows the user to save the white reference measurement to the RAM of the built-in computer. HCFR is computed
for subsequent target reflectance measurements based on the saved reference. This is saved to the output file,
eliminating the need to calibrate the target measurements to the white reference in post-processing. A new SRT-- 99-020
Spectralon (serial number 99AA08-0918-1593) manufactured by Lab Sphere was used for the measurement of the
white reference. The ASD data files were imported into a python script for further analysis using the Python module
SpecDal (Lee, 2017) to read the ASD format. Further data analysis was carried out using numerous other Python (Van
Rossum and Drake., 2009) packages, mainly NumPy (van der Walt et al., 2011), pandas (McKinney, 2010), Matplotlib
(Hunter, 2007), Rasterio (Gillies et al., 2013), GeoPandas (GeoPandas developers, 2019), rasterstats (Perry, 2015), and
PyEphem (Rhodes, 2020).
**2.3 Satellite data**
We compare the in situ measurements with surface reflectance products derived from a Landsat-8 Operational Land
Imager (OLI) scene acquired on September 3rd, 2019 (10:10 GMT), the day before the field campaign, and a Sentinel
2A scene acquired on September 4th (10:20 GMT), the same day as the field campaign. Both scenes are cloud free over
the study area (Figure 2). Details on the atmospheric correction algorithm used to generate the Landsat-8 OLI level-2
surface reflectance data product from top of atmosphere (TOA) reflectance can be found in Vermote et al. (2016) and in
the product guide of the algorithm used to derive surface reflectance (USGS, 2020). Details on the equivalent Sentinel-2
product – the Level-2A bottom of atmosphere reflectance – are given in Main-Knorn et al. (2017) and Richter and
Schläpfer (2011). For the sake of readability, we refer to the Landsat-8 OLI level-2 surface reflectance as "Landsat"
data in the following, and to the Sentinel-2 level-2A surface reflectance as "Sentinel" data. The Landsat and Sentinel
surface reflectance raster data used in this study were acquired using Google Earth Engine (Gorelick et al., 2017).
The wavelength range of the spectroradiometric measurements carried out on the ground overlaps with bands 1-5 of the
Landsat data and bands 1-9 and 8A of the Sentinel data, respectively. Only spectral ranges covered by these bands are
considered for this study. The wavelengths and resolution of the individual bands, as well as the relevant viewing and
solar angles are given in Table 3. For each ground measurement point, band values were extracted from the satellite
scenes at the overlaying pixel.
In order to compare the satellite values with ground data, we compute mean values for the subsets of the spectral
reflectance curves measured on the ground that correspond to the Landsat and Sentinel bands, respectively. Data are
then grouped into profile lines and/or different bands, the Pearson correlation is computed for ground- and
corresponding satellite data, and further comparisons are carried out using standard statistical metrics.
To assess the influence of the spatial resolution of the satellite data on results, band 3 imagery was resampled (cubic
interpolation) from the original 10 m resolution to 30 m and 60 m for Sentinel and from 30 m to 60 m for Landsat,
respectively. To account for the potential effects of the uncertainty in the GPS coordinates, we created a circular buffer
with a radius of 3m around each in situ measurement point. For each buffer, the corresponding satellite value is
computed as the median of the values of all pixels the buffer overlaps with.

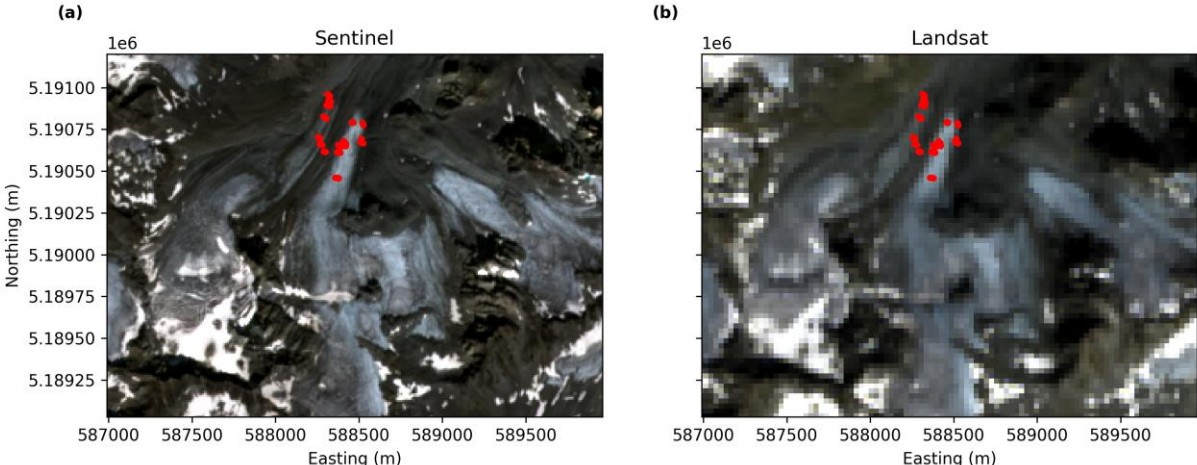

Figure 2: Jamtalferner as seen in the Sentinel (a) and Landsat (b) scenes used in this study. The images shown here are composites of bands 2, 3, and 4 of each satellite's L2A surface reflectance product displayed at a resolution of 10 (Sentinel) and 30 (Landsat) m/pixel, respectively. Profiles where reflectance spectra were collected are marked in red. Coordinate reference system: EPSG: 32632.

## 3. Results

### 3.1 Surface measurements

The in situ measurements exhibit extreme differences in HCFR depending on the characteristics of the surface. Figure 3 shows the spectra grouped into profiles, with the mean spectral HCFR highlighted for each profile. P3 is the "brightest" profile, with the highest maximum (up to 0.7) and minimum (up to 0.2) values of all profiles. Profiles 2, 11, and 14 are the darkest profiles and all of their respective spectral reflectance remain below 0.2 at all measured wavelengths. Figure 3 also shows the ice surface along profile lines 3 (brightest) and 11 (darkest) for a visual comparison. In P3, the surface is mainly comprised of clean, dry ice. In P11, the ice surface is wet and impurities (rocks, fine grained debris) are present. The profile line crosses several small melt water channels with running water.

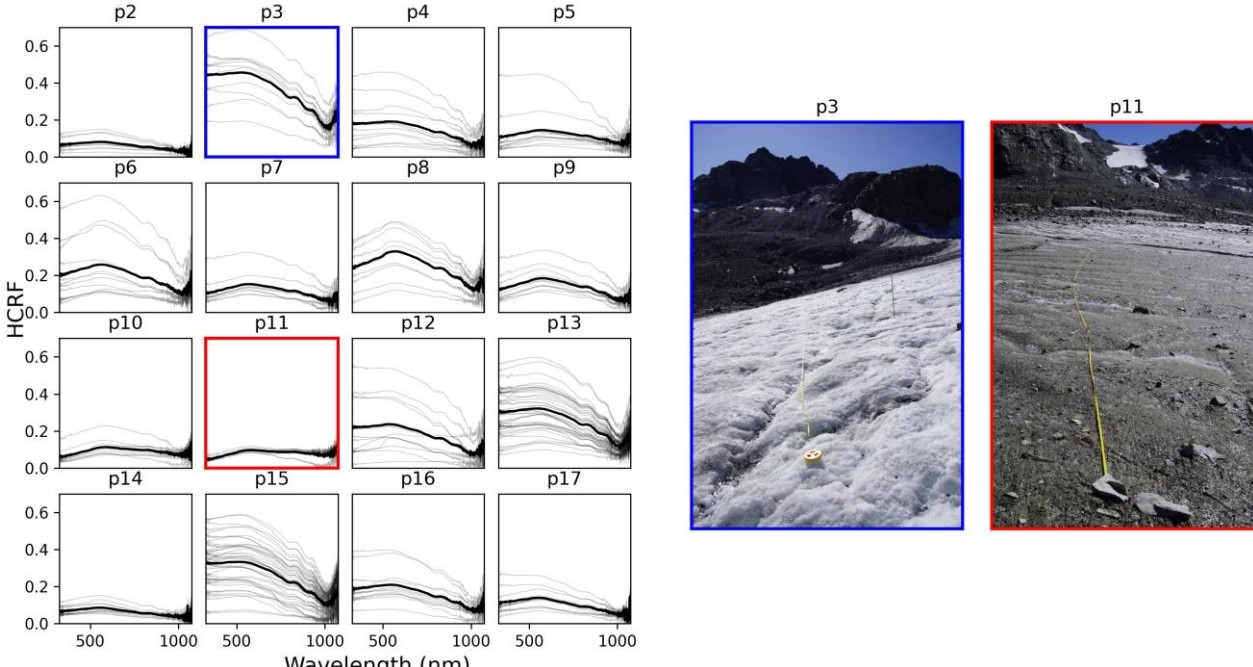

Figure 3: Each subplot on the left shows the spectra along a profile line. The bold black lines highlight the mean spectral reflectance (HCRF) in each profile. Fotos of the ice surface along p3 and p11 are shown on the right for visual context. Fotos were taken at the time of the respective measurements by A. Fischer.

Table 2 contains a qualitative description of the ice surface along each profile line, the length of the line, the number of spectra per line, and the number of Landsat and Sentinel band 3 pixels that each line crosses, as well as the mean solar elevation and azimuth angles for the profile. The maximum number of pixels per line is 5 for Sentinel and 3 for Landsat, respectively. All lines cross at least 2 pixels for Sentinel, while 3 lines fall into a single Landsat pixel. See Fig. 1 for the location of each profile on the glacier.

Table 2: Description of the surface characteristics along each profile line, as well as number of spectra collected along the line and number of pixels intersected by the line in band 3 of the Sentinel and Landsat scenes, respectively.

| Profile Nr. | Qualitative description | Mean solar elevation, azimuth in degrees | Spectra | Sentinel B3 pixels | Landsat B3 pixels |
|---|---|---|---|---|---|
| P2 | Relatively smooth, uniform ice surface, slightly wet. | 24.69, 106.70 | 11 | 3 | 2 |
| P3 | Mostly dry surface, clean cryoconite. | 26.43, 108.92 | 11 | 4 | 1 |
| P4 | Mostly dry ice surface, some dirt, some rocks/debris on ice surface where profile approaches moraine. | 28.64, 111.87 | 11 | 4 | 2 |
| P5 | Significant debris cover along profile. Where ice is exposed, ice surface is wet. Profile crosses meltwater channels with running water. | 31.34, 115.72 | 11 | 3 | 1 |
| P6 | Wet ice surface with dust/dirt transitions to cleaner, brighter ice. | 34.45, 120.61 | 11 | 4 | 1 |
| P7 | Grey-ish ice surface with meltwater channels and fine-grained debris/small rocks. | 36.20, 123.57 | 11 | 2 | 2 |
| P8 | Similar to P7, fewer rocks. | 38.05, 126.99 | 11 | 4 | 2 |
| P9 | Wet ice surface with mixture of relatively clean cryoconite and more dusty areas. | 39.40, 129.68 | 11 | 3 | 2 |
| P10 | Wet ice surface with several small melt water channels. Mostly dirty, grey ice. | 40.71, 132.51 | 11 | 3 | 2 |
| P11 | Wet ice surface with several small meltwater channels. Very dirty ice with scattered small rocks. | 42.08, 135.75 | 11 | 4 | 2 |

| | | | | | |
|---|---|---|---|---|---|
| **P12** | Relatively clean, bright ice interspersed with larger meltwater ponds/channels, which contain dirt and small rocks. | 47.61, 153.83 | 11 | 4 | 3 |
| **P13** | Clean cryoconite with some darker patches. | 48.63, 159.14 | 40 | 5 | 2 |
| **P14** | Wet ice surface with fine grained dirt in relatively uniform cryoconite. | 49.80, 168.30 | 11 | 4 | 2 |
| **P15** | Uneven ice surface, mostly clean, dry ice. | 50.30, 179.29 | 40 | 3 | 2 |
| **P16** | Mixture of wet and dry ice surface and fine-grained dirt. | 49.43, 194.99 | 11 | 3 | 2 |
| **P17** | Mostly wet ice surface, fine grained dirt with some cleaner patches. | 48.33, 202.34 | 11 | 2 | 2 |

The spectral reflectance curves of the individual spectra as well as of the profile lines indicate high spatial variation of
surface types and associated reflective properties. The spectral signatures of the individual spectra can roughly be
grouped into dry ice, wet ice, and dirt/rocks. (We use the word "dirt" to describe all types of mineral or organic
materials and fine-grained debris that may collect on the glacier surface.) However, transitions between these types are
gradational and in practice these categories cannot always be clearly separated - both dry and wet ice might be clean or
dirty, dirt might be wet or dry.
The reflectance curves for clean ice exhibit the typical shape frequently found in literature (Zeng et al., 1984), with
highest reflectance values (up to 0.69) in the lower third of our wavelength range and declining values for wavelengths
greater than approximately 580 nm. The spectral reflectance curves of wet ice surfaces follow roughly the same shape
as for dry ice but are strongly dampened in amplitude with reflectance values typically not exceeding 0.2. In contrast,
the reflectance curve of dirty surfaces remains at uniformly low values throughout our wavelength range in some cases
and exhibits an increase between 325 and approximately 550 nm before flattening out in other cases. Reflectance values
have similar magnitudes as for wet ice. Example reflectance curves of these surface types are given in Fig. 4.

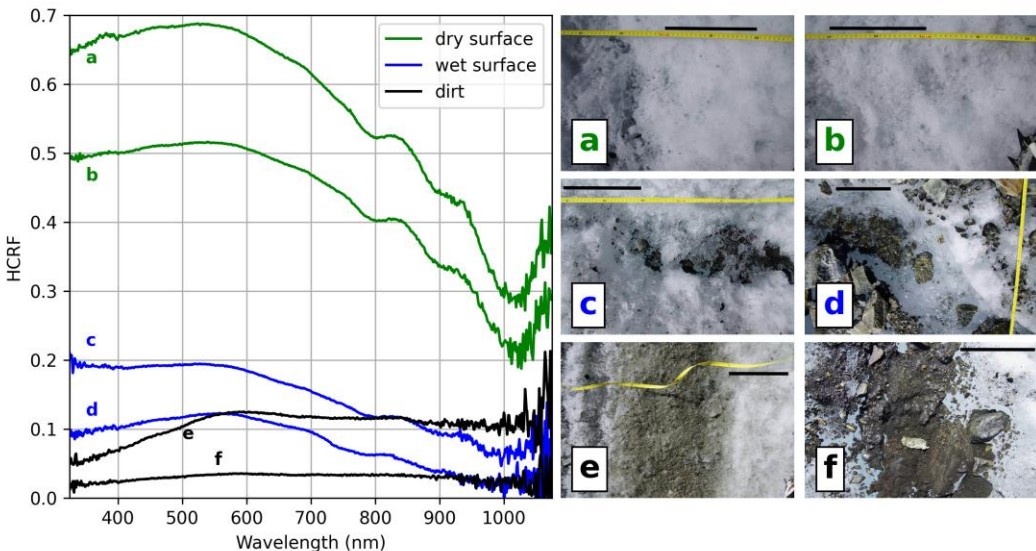

Figure 4: Spectra of different kinds of ice surface types encountered in the ablation zone of Jamtalferner. The photos on
the right show the ice surface at the sampling sites of the respective spectra. The black bar in each photo represents
approximately 20 cm, to provide a sense of scale. The spectra shown in this figure are part of the following profile lines:
a, b, c – p3; d – p4; e – p6; f – p12.
**3.2 Comparison with satellite data**
Figure 5 shows all measured spectral reflectance curves, as well as the Sentinel and Landsat values in the bands that
overlap the wavelength range of the ground measurements. Reflectance values were extracted from the satellite imagery
at the coordinates of each sampling point and overlayed onto the plots of the in situ spectra as coloured bars. Naturally,
neither satellite captures the full range of reflectance values measured on the ground. In all overlapping bands of
Sentinel and Landsat, the Sentinel values are higher, in the sense that the maximum values of the Sentinel data are
closer to the maximum values measured on the ground, while the minimum Landsat data are closer to the minimum
values measured on the ground.

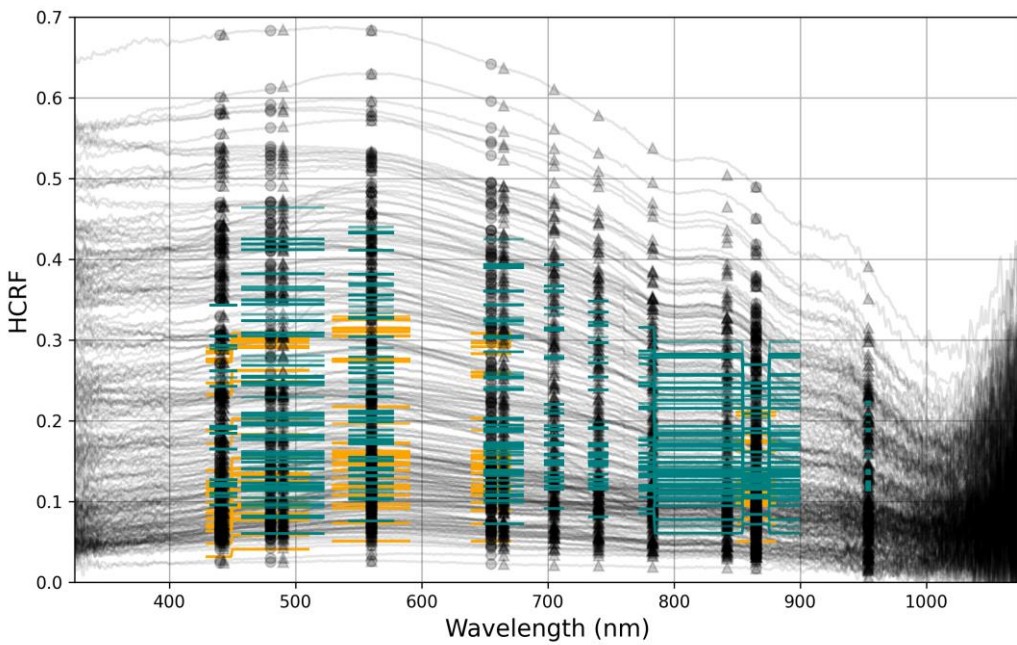

Figure 5: The spectra measured in situ are plotted in black. Black circles indicate the central wavelengths of the Landsat
bands, black triangles those of the Sentinel bands (see Table 3). Orange and blue lines represent the wavelength range of
the respective Landsat and Sentinel bands along the horizontal axis and the satellite derived reflectance at the sampling
points of each spectrum on the vertical axis.
Comparing the mean of the HCRF spectra measured on the ground for each satellite band with the associated satellite
values yields a Pearson correlation coefficient ranging from 0.53 (band 5) to 0.62 (band 1) for the Landsat bands and
0.3 (band 9) to 0.65 (band 2) for Sentinel. Table 3 lists the correlation coefficients, as well as the wavelength range and
resolution of each band. The two lower resolution Sentinel bands (band 1, band 9 – 60m resolution) have notably lower
correlation coefficients than the higher resolution bands. The Sentinel and Landsat data at the in situ measurement
points are strongly correlated with each other in the bands where both satellites overlap, with r=0.69 in band 1 and r>0.8
for bands 2, 3, 4, and 5.
For a visual comparison of the location of the profile lines and the range of measured values in the profiles in relation to
the satellite pixel boundaries and pixel band values, see Fig. 6 for Sentinel (band 3 selected as an example) and
supplementary material for an analogous figure of the Landsat data.
Table 3: Band names and respective wavelength range and resolution for Landsat and Sentinel as used in this study.
Pearson correlation given for mean band values of ground-measurements and associated satellite data. For Landsat, the
solar zenith and azimuth angles given in the surface reflectance image are listed. The view zenith angle is hardcoded to
0 in the Land Surface Reflectance Code (LaSRC_1.3.0) for the Landsat surface reflectance product, as per the LaSRC
documentation (USGS, 2020). For Sentinel, the incidence angles refer to the mean viewing zenith and azimuth angles
for each band. The solar angles are the averages for all bands.

| Band | Landsat | Sensing time: 2019-09-03 10:10 GMT | | | | | |
|---|---|---|---|---|---|---|---|
| Band | Range (nm) | Resolution (m) | Pearson Corr. | View zenith angle | View azimuth angle | Solar zenith angle | Solar azimuth angle |

| | | | | | | | |
|---|---|---|---|---|---|---|---|
| 1 (Coastal/Aerosol) | 430-450 | 30 | 0.62 | 0 | - | 42.63 | 153.57 |
| 2 (Blue) | 450-510 | 30 | 0.61 | | | | |
| 3 (Green) | 530-590 | 30 | 0.58 | | | | |
| 4 (Red) | 640-670 | 30 | 0.57 | | | | |
| 5 (NIR) | 850-880 | 30 | 0.53 | | | | |
| | **Sentinel** | Sensing time:2019-09-04 10:20 GMT | | **Mean incidence zenith angle** | **Mean incidence azimuth angle** | **Mean solar zenith angle** | **Mean solar azimuth angle** |
| 1 (Coastal/Aerosol) | 433-453 | 60 | 0.46 | 3.13 | 193.02 | 40.83 | 159.93 |
| 2 (Blue) | 457.5-522.5 | 10 | 0.65 | 2.48 | 198.51 | | |
| 3 (Green) | 542.5-577.5 | 10 | 0.63 | 2.59 | 196.22 | | |
| 4 (Red) | 650-680 | 10 | 0.61 | 2.72 | 194.92 | | |
| 5 (Vegetation Red Edge) | 697.5-712.5 | 20 | 0.57 | 2.79 | 194.43 | | |
| 6 (Vegetation Red Edge) | 732.5-747.5 | 20 | 0.56 | 2.87 | 193.84 | | |
| 7 (Vegetation Red Edge) | 773-793 | 20 | 0.55 | 2.95 | 193.53 | | |
| 8 (NIR) | 784.5-899.5 | 10 | 0.56 | 2.54 | 197.22 | | |
| 8A (NIR narrow band) | 855-875 | 20 | 0.53 | 3.04 | 193.30 | | |
| 9 (Water vapour) | 953-955 | 60 | 0.3 | 3.22 | 192.89 | | |


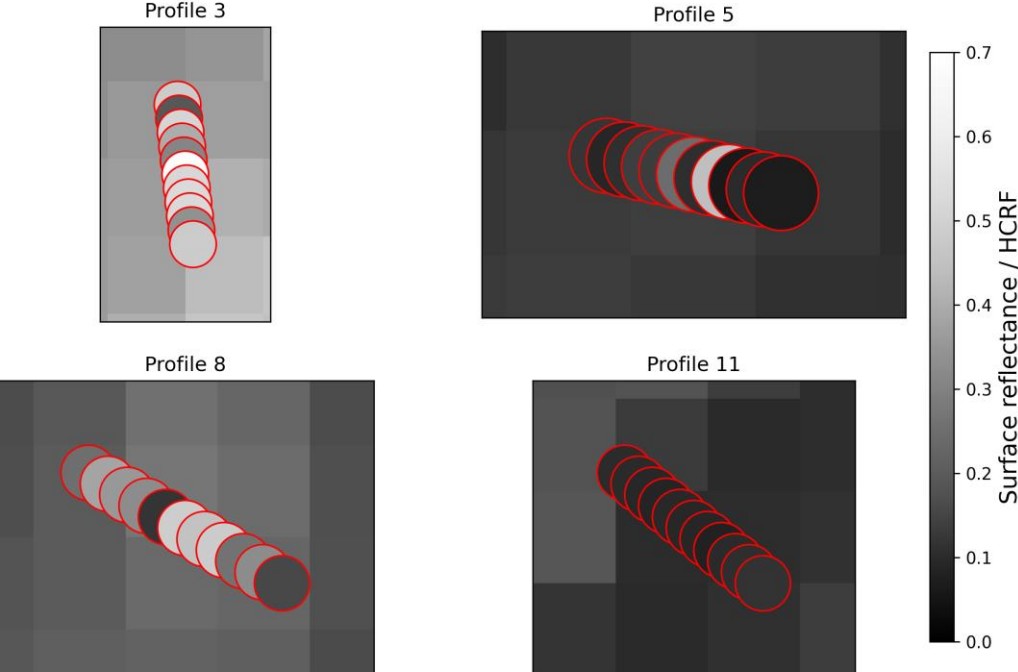

Figure 6: The spectra comprising the profile lines are plotted over the corresponding satellite pixels for selected profiles.
The colour bar is the same for the background raster and the circles indicating the sampling sites of the spectra and
represents the Sentinel band 3 pixel value and the mean reflectance in the Sentinel band 3 wavelength range of each
spectrum, respectively. The pixel size of the raster is 10m². The GPS coordinates of the sampling sites are centred in the
circles. The circle radius is set to 3m to represent the horizontal uncertainty of the GPS points.
The spread of in situ HCRF values per profile is generally lower for profiles that are darker overall, and greater for
brighter profiles, although not in all cases (Fig. 3, Fig. 7). In the Sentinel band 3 wavelength range, profile 3 is brightest
with a median reflectance of 0.48 and spread of 0.49. Profile 6 (median in Sentinel band 3 range: 0.21) has the largest
spread of HCRF (0.52). Broadly speaking, profiles with a high median HCRF tend to include individual measurement
points that are both very bright and very dark, while darker profiles are more uniformly dark. Profile 6 in particular
transitions between surface types and contains wet/dirty spectra as well as dry ice spectra (see Table 2). Figure 7 shows
boxplots of the ground measurements (band 3 mean) for all profiles to exemplify this and indicates where the Landsat
and Sentinel values fall compared to the spread of values in each profile.

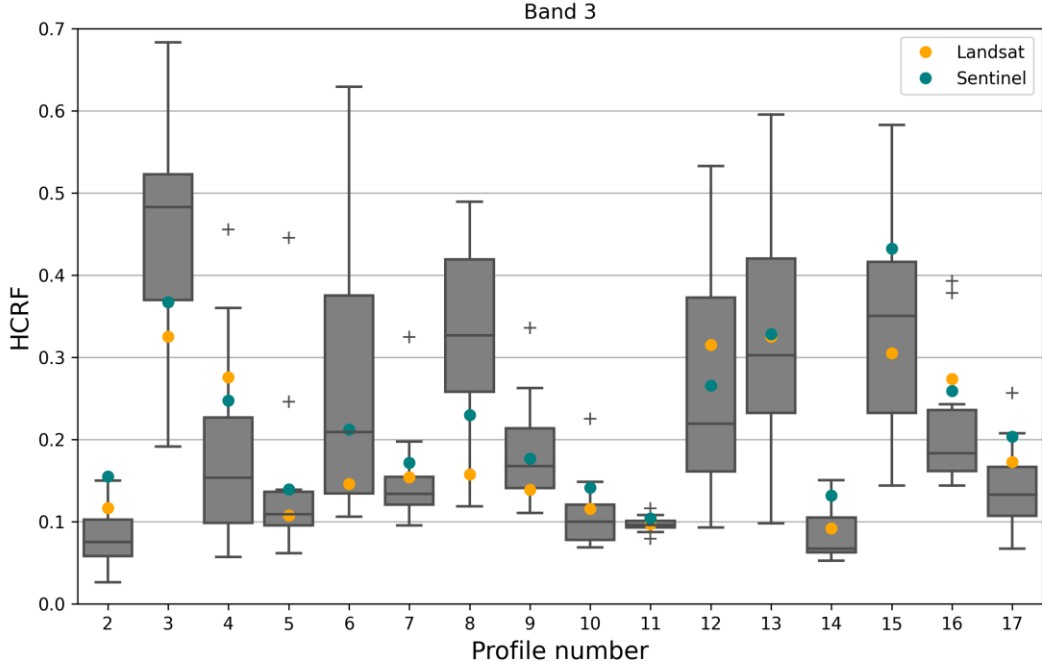

Figure 7: Spread of the Sentinel band 3 (wavelength range: 542.5-577.5 nm) mean values of the measured spectra,
grouped by profile. Orange and blue circles show corresponding mean pixel values of data extracted from Landsat and
Sentinel pixels at the sampling sites of the spectra, respectively. The boxes represent the first and third quartile. The
whiskers represent 1.5x the interquartile range, the + symbols are outliers.
When binning in situ measurements by the associated satellite value/pixel and taking the median or mean of the binned
values, the difference between the median/mean in situ value and the satellite value tends to decrease with increasing
number of in situ measurements mapped to unique satellite values. This is to be expected, as each satellite value
represents an integration of the emission characteristics over the area contained in the pixel. However, for our data, this
relationship is not obviously linear and differs between Sentinel and Landsat, as well as between different bands (Figure
361 8).

Comparing in situ and satellite values for individual in situ measurement points, it is apparent that both satellites tend to
overestimate the reflectance values of dark ground surfaces, and underestimate the reflectance of bright surfaces, in all
bands (Figure 9). The shift from over- to underestimation appears linear and has a similar increase rate in all bands. The
zero crossings of the regression lines, i.e. the ground reflectance values for which ground measurements and satellite
values match, fall between 0.15 (band 5) and 0.21 (band 1) for Landsat and 0.17 (band 9) and 0.27 (band 3) for
Sentinel.
Figure 10 shows histograms of the mean reflectance in band 3 of Landsat and Sentinel, respectively, compared with
associated in situ values, as well as density plots of the satellite derived surface reflectance over all pixels in the study
area. The mean is highest in the in situ measurements and lowest in Landsat images. Both Sentinel and Landsat fail to
capture HCRF values below 0.05 and above 0.45. A second peak in frequency evident from the in situ measurements at
a reflectance of 0.4 is not represented in the remote sensing data.

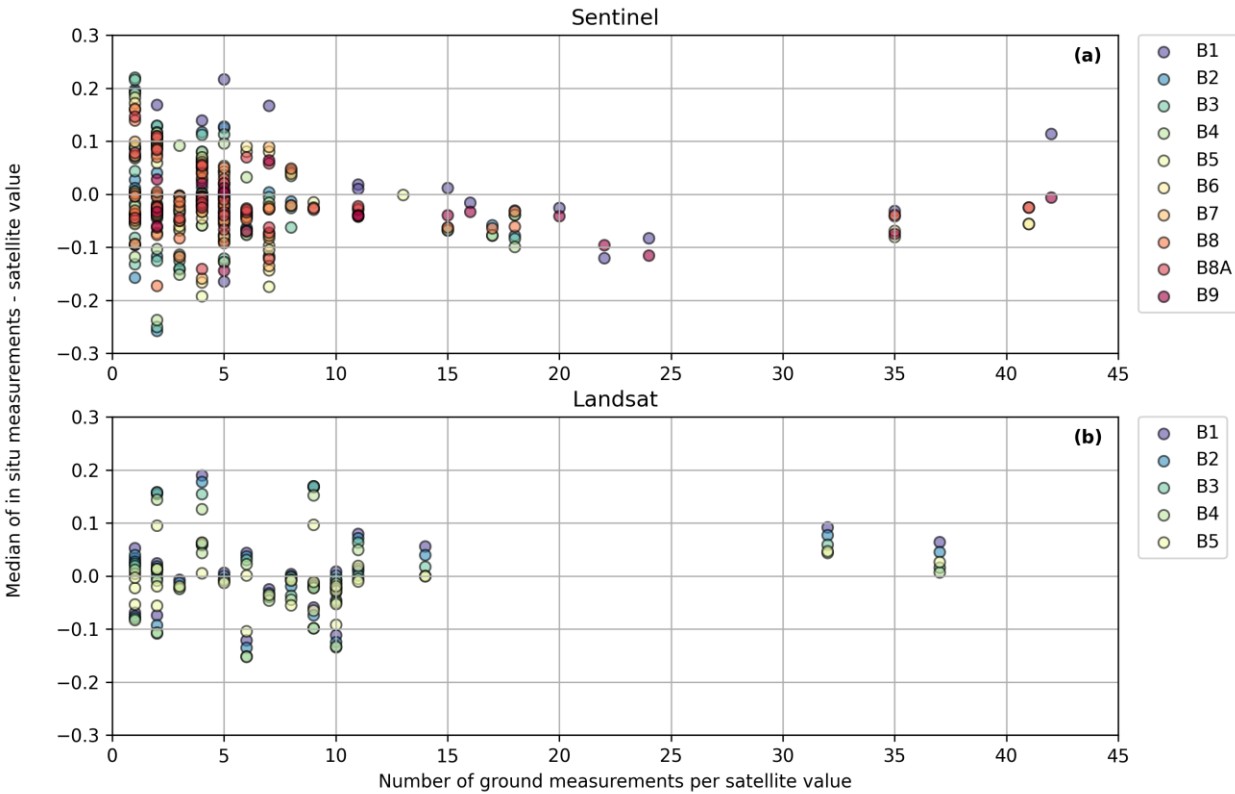

Figure 8: The number of ground measurements per unique satellite value (x-axis) is plotted against the difference between the median of these ground measurements in the respective wavelength band and the corresponding satellite value (y-axis). i.e. values that are positive in the vertical axis represent cases where ground reflectance is higher than satellite derived reflectance, whereas negative values represent the opposite. Different colours represent the different satellite bands, as indicated by the legend next to the plots.

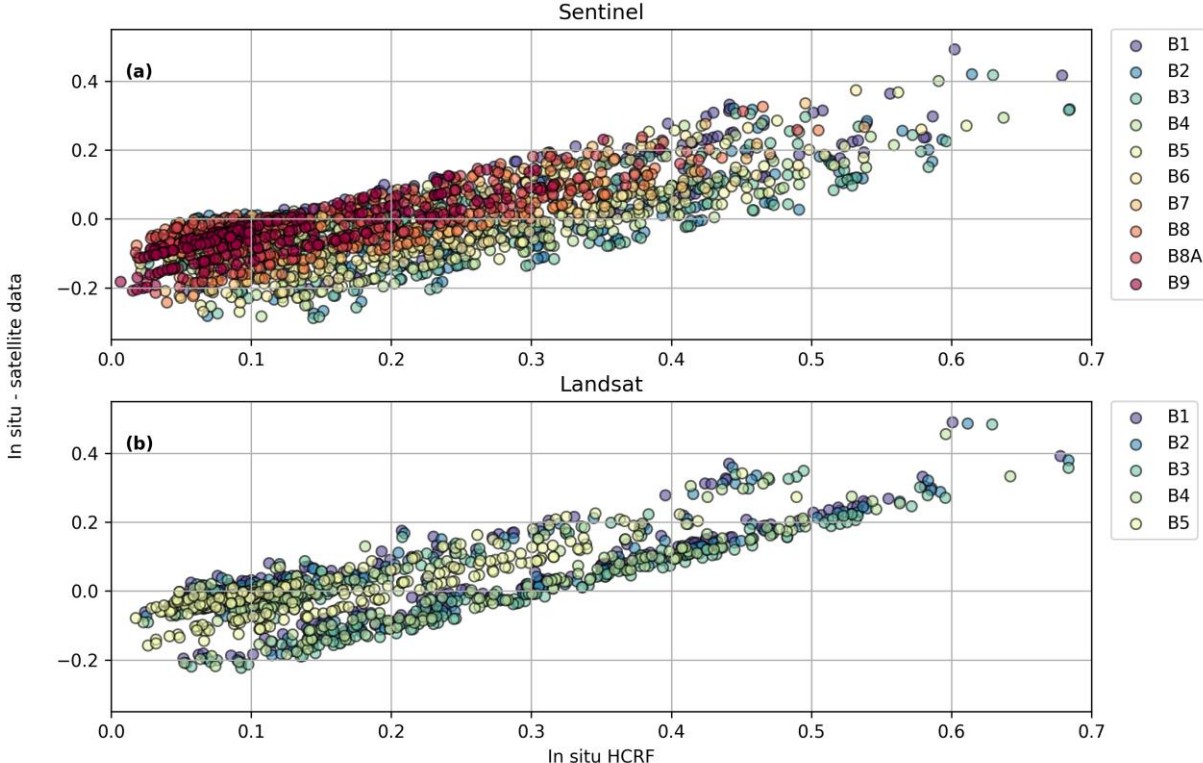

Figure 9: Same data as in Fig. 8, but showing individual sampling points without grouping by common satellite pixels.

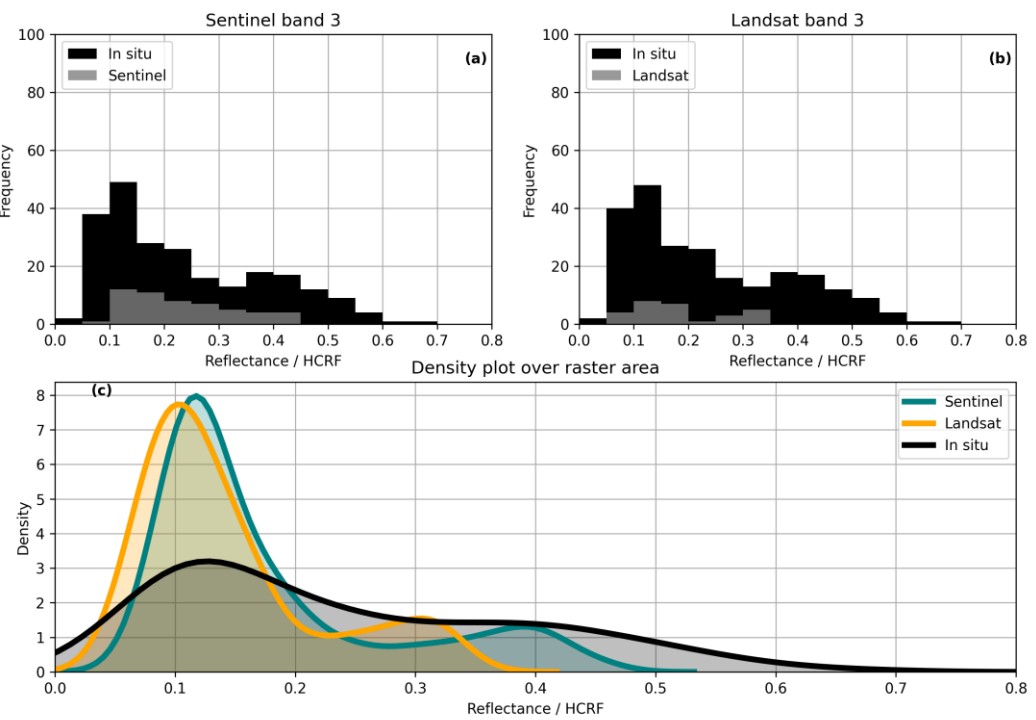


Figure 10: The histograms in the top panels (a, b) show the frequency of occurrence of the band 3 mean values of the
ground measurements per reflectance bin. Bin width: 0.05. Overlaid in grey are the histograms of the corresponding
satellite pixel values. The bottom panel (c) shows density plots of the Sentinel and Landsat band 3 surface reflectance
rasters over the study area (smallest possible rectangle containing all ground measurements), with the density of the in
situ HCRF for comparison.
To conclude the results, a note on the sensitivity of data and results to the spatial resolution of the satellite data and the
accuracy of the geolocation of the in situ data: To assess the possible effects of the GPS accuracy or lack thereof, we
compare the differences between in situ and satellite values presented previously to the differences that result when a
buffer corresponding to the GPS uncertainty is created around each in situ measurement point. For the Sentinel data in
the original 10 m resolution of band 3, the maximum number of pixels that any buffer touches is 4, the mean is 2.6, and
most buffered in situ measurement points overlap with 2 pixels. For the 30 m Landsat data in band 3, the maximum
number of pixels touched is also 4 while the mean is 1.5 and most in situ points are fully within only one pixel. Table 4
gives the standard deviation of differences between the in situ HCFR and the satellite data in different resolutions,
grouped by the number pixels the buffered measurement points overlap with, to show how variability of results shifts
depending on the buffer and the raster resolution. Changes caused by introducing the buffer are small in all groups. As
expected, standard deviation increases with decreasing resolution of the satellite pixels due to the loss of detail in the
satellite data. Figure 11 gives an overview of the ungrouped dataset with and without the buffer and at different raster
resolutions.
Table 4: Comparison of in situ and satellite data by the standard deviation (SD) of the difference between in situ HCFR
and satellite surface reflectance. Values are grouped by number of pixels that buffered in situ measurements overlap.

| | Sentinel | | | | | Landsat | | | |
|---|---|---|---|---|---|---|---|---|---|
| Nr. of overlapping pixels | Nr. of points | SD, no buffer, 10m | SD, buffer, 10m | SD, buffer, 30m | SD, buffer, 60m | Nr. of points | SD, no buffer, 30m | SD, buffer, 30m | SD, buffer, 60m |
| 1 | 25 | 0.098 | 0.098 | 0.108 | 0.129 | 134 | 0.129 | 0.129 | 0.134 |
| 2 | 124 | 0.119 | 0.118 | 0.120 | 0.124 | 94 | 0.106 | 0.107 | 0.103 |
| 3 | 9 | 0.057 | 0.065 | 0.069 | 0.099 | 1 | - | - | - |
| 4 | 76 | 0.122 | 0.121 | 0.127 | 0.136 | 5 | 0.082 | 0.074 | 0.083 |


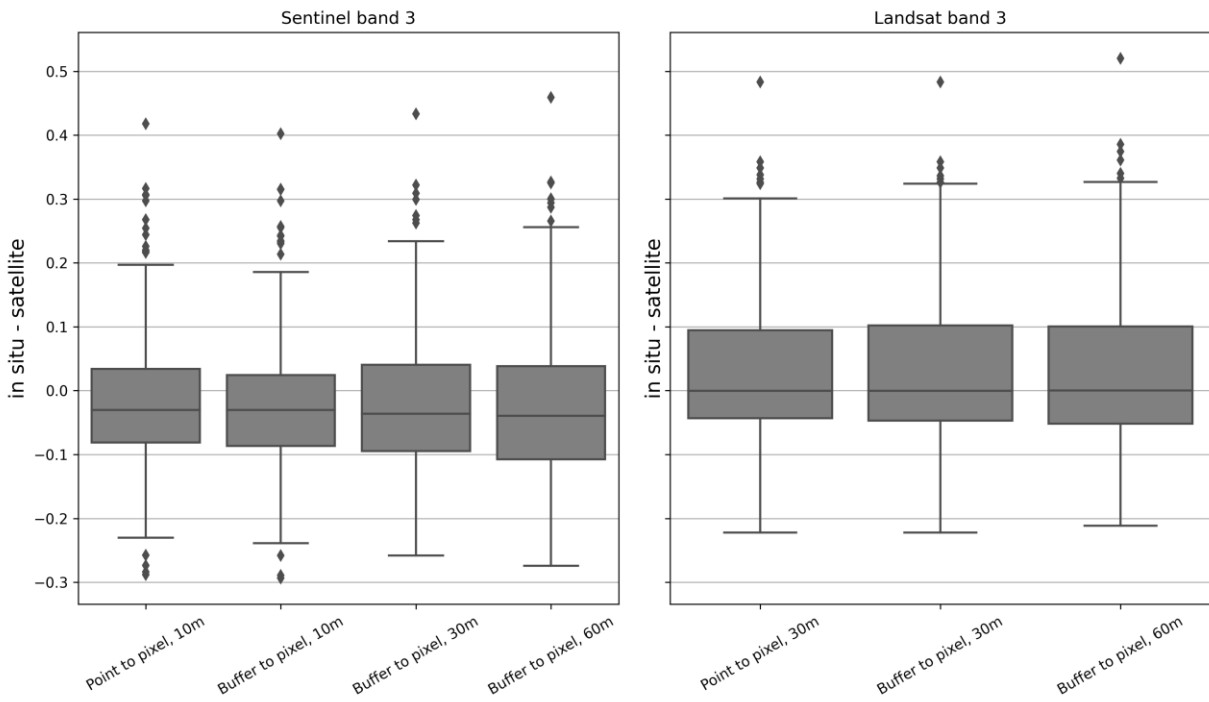

Figure 11: For the respective Sentinel and Landsat band 3 wavelength range, the difference between the in situ HCRF
and satellite surface reflectance product is on the vertical axis. Point to pixel refers to the data as presented in previous
figures. Buffer to pixel refers to data generated using a buffer around the in situ measurement points to account for GPS
accuracy. For Sentinel, the original 10m resolution data was resampled to 30 and 60m. For Landsat, the original 30m
resolution data was resampled to 60m.
**4. Discussion**
There are a number of complexities associated both with measuring reflectance properties on the ground and with any
comparison between different products and data sets. Perhaps more than anything else, our results highlight the need for
further in situ measurements and targeted data collection campaigns designed specifically to address some of the
uncertainties detailed in the following.
**4.1. Reflectance anisotropy and changing solar and atmospheric conditions**
Ice is an anisotropic material and previous studies have shown that for glacier surfaces, anisotropy increases with
decreasing albedo and depends on wavelength and solar zenith angle (Greuell and de Wildt, 1999; Klok et al., 2003;
Naegeli et al., 2015). In order to truly quantify the effects of anisotropy in in situ spectroradiometric measurements, the
bidirectional reflectance distribution function (BRDF) must be obtained – ideally for each measurement point. The
BRDF cannot be measured directly but is approximated, e.g. by interpolating between multi-angular spectroradiometer
measurements (Naegeli et al., 2015), or with modelling approaches (Malinka et al., 2016). While multi-angular HCRF
measurements allow for the estimation of the BRDF, they are intrinsically dependent on the atmospheric conditions
(cloud cover) at any given time, as well as on the topography and structure of the surface. Naegeli et al. (2015, 2017)
use this approach to develop anisotropy correction factors for different glacier surface types in order to account for the
typical underestimation of albedo in observations from nadir in remote sensing data. They find a difference between
corrected and uncorrected albedo values of up to 11% for dirty ice in airborne imaging spectroscopy data. Nonetheless,
the application of constant correction factors for clustered surface types is a simplification that obscures both the
gradational nature of surface classification and the complexity of accounting for the effects of varying surface
roughness on effective illumination angles. We consider a quantitative assessment of anisotropy beyond the scope of our
study and hope to tackle this issue in detail in future work. We assume that our in situ data as well as the satellite
products underestimate the quantities they measure (HCRF and surface reflectance as per the respective documentation
of the satellite products) due to the nadir or near-nadir observational angle, in particular for dark surfaces, and that
uncertainties caused by anisotropy are likely to be in the range found by Naegeli et al. (2017). The local variability of
reflectance properties of glacier ice is comprised of the spectral, as well as spatial and temporal variability of reflectance
anisotropy, which require a combination of targeted, continuous measurements and modelling that accounts for the
surface roughness of different glacier surface types to truly delineate.
The weather on September 3 (Landsat overpass) and September 4, 2019, (Sentinel overpass, in situ measurements) was
very favourable. There was no cloud cover at the study site during either of the satellite overpasses and for the duration
of the field measurements and we consider any changes in atmospheric conditions to be negligible. While the
illumination angles naturally change over the course of the day and accordingly changed during the in situ
measurements (Table 2), very low solar elevation angles were avoided. In their study on parametrizing BRDFs for
glacier ice and Landsat TM, Greuell and de Wildt (1999) show that the spectrally integrated albedo of dark ice changes
with the solar zenith angle and is particularly low for low zenith angles. Accordingly, we acknowledge that the changing
solar angles are a source of uncertainty in our data and the comparison with the satellite derived reflectances, but we
consider this uncertainty relatively small since measurements were carried out within a few hours before and after the
satellite overpasses, avoiding very low solar elevation angles. Greuell and de Wildt (1999) also point out that the drop in
albedo for low zenith angle is related to the presence of meltwater at later times of day (lower zenith angles), which
highlights the difficulty of isolating one variable (zenith angle) in a complex system with multiple variables that change
over time (surface processes like meltwater affecting reflectance properties).
The Landsat and Sentinel surface reflectance products both incorporate an atmospheric correction applied to TOA
reflectance in the generation of the BOA product (Vermote et al., 2016; Main-Knorn et al., 2017). This introduces some
uncertainty into the comparison with in situ data since the correction methods differ. Nonetheless, we believe that
assessing how in situ data compare to the frequently used surface reflectance products of the Landsat-8 and Sentinel-2
suites is a necessary first step in being able to determine whether custom atmospheric corrections would improve results
and if such improvements would be large enough to outweigh the added complexity and computational cost. We suggest
that the answer to this question depends on the application and the spatial scale of the intended analysis. Again, this is
beyond the scope of the presented study and is a point that needs to be specifically addressed in future work. We suggest
that case studies at individual, well-studied glaciers can serve as an ideal testing ground for such issues, and will help to
determine whether custom atmospheric corrections should be applied and are feasible on a regional or even global scale
in satellite-based studies of ablation area reflectance properties.
**4.2. Implications of in situ and satellite comparison**
The results presented in section 3.1. highlight the large spatial variability of HCRF and different surface types
encountered in the ablation area, both of which are in line with findings from other studies (Naegeli et al., 2015, 2017;
Di Mauro et al., 2017). Section 3.2., the comparison of the in situ data with satellite values, arguably presents greater
challenges in terms of interpretation and implications of the results.
In summary, there are three key findings which we believe may be important for further studies and for delineating the
relationship between in situ and satellite derived reflectance:
• Sentinel surface reflectance values tend to be closer to the higher end of HCFR values measured in situ, while
Landsat tends to be closer to the in situ minimum.
• The difference between in situ data and satellite data tends to decrease when there are more in situ data points
per pixel, but not always and not in a clearly linear way.
• The reflectance of dark surfaces tends to be overestimated in the satellite products, while the reflectance of
bright surfaces tends to be underestimated.
Explaining the above points in full requires targeted investigations specifically addressing the contributing factors and
uncertainties, which – with our current data set – we can only provide a qualitative overview of:
As mentioned previously, different atmospheric corrections are used for the Sentinel and Landsat surface reflectance
products. This may contribute to systematic differences in how surface reflectance is represented under differing
lighting conditions and in different spectral ranges. Efforts to harmonize the Landsat and Sentinel surface reflectance
data sets have great potential for minimizing this problem for applications where data from both satellites is used
(Claverie et al., 2018).
Another issue that deserves more detailed attention is the narrow/spectral to broad band conversion required for
comparing satellite reflectance in individual bands with the in situ data of the same wavelength range. We intentionally
do not compute a shortwave broadband albedo from the satellite band values or the spectral in situ data to avoid
introducing a further source of uncertainty. Instead, we limit ourselves to averaging over the band wavelength range in
order to keep the comparison as straightforward as possible, but acknowledge that a glacier wide broad band albedo is a
key parameter for many regional or global modelling applications.
The standard atmospherically corrected BOA reflectance products from satellite data are provided without correcting
for the BRDF. The BRDF, describing the change of the reflectance with different observation and incidence geometries,
can have a significant impact on the satellite-based reflectance as well as on the in situ data, leading to inherent
challenges when comparing satellite based BOA reflectance with in situ reflectance measurements (Schaepman-Strub et
al., 2006). Correcting Landsat and Sentinel surface reflectance with MODIS or VIIRS BRDF products to produce
surface albedo has been shown to be a viable approach in some cases (Shuai et al., 2011; Li et al., 2018), but the coarse
resolution of MODIS and VIIRS data is unlikely to capture the small-scale anisotropy effects of different glacier surface
types. This would therefore be of limited use for our purposes. Optimizing methods for computing surface albedo from
the L2A products, as well as from the in situ HCRF, requires further study and customized solutions accounting for
local topographic effects and the spectral characteristics of the surfaces. We assume that for our case uncertainties due
to the intrinsic difference between HCRF and satellite derived HDRF are small compared to other sources of
uncertainty: The influence of local topography as a source of indirect radiation is not represented in the satellite derived
values and the microstructure of the ice surface may locally affect in situ values on a scale that not visible to the
satellite, but could be very significant for in situ measurements (e.g. small ice ridges or similar features acting as
reflectors and/or scattering light into the FOV of the instrument).
Hendricks at al. (2004) state for spectroradiometric measurements at Hintereisferner compared to Landsat ETM+
imagery acquired about 2 weeks before the field measurements: "The reflectance of ice seems to be highly variable with
both under -and overestimations of up to 76 % and 31 % respectively." This corresponds well with our finding that both
under- and overestimation occur frequently for both satellites. The factors mentioned above may partly explain the
location of the shift from under- to overestimation (Fig. 9), but –again- targeted measurement campaigns are needed to
truly quantify this.
The influence of very local backscattering could play a role in the seeming inconsistencies in the dependency of the
difference between in situ and satellite data the on number of in situ measurement points per pixel (Fig. 8), but this also
ties in with questions regarding the positional accuracy of the in situ measurement points and the satellite data, and the
spatial representativity of point measurements for a larger area:
Our comparison of in situ and satellite data is based on the assumption that we know where both are located in a
common coordinate reference system to a sufficient degree of accuracy. The accuracy of the position of the GPS points
at the start and end points of the measurement profiles is approximately 3m. Sentinel-2 orthorectification is based on the
PlanetDEM 90 digital evelation model (DEM), which incorporates the SRTM DEM in areas where SRTM is available,
such as Austria (Kääb et al. 2016). The geometric accuracy of the Sentinel data hence depends on the accuracy of the
underlying DEM, which is subject to a number of uncertainties particularly over mountainous terrain. Vertical
inaccuracies – which propagate into horizontal inaccuracies - increase over glacier surfaces, especially in areas with
large changes in surface elevation, as the DEM can only provide a snapshot of conditions for a moment in time and
quickly becomes outdated in rapidly changing environments. Pandžic et al (2016) determine an average offset in the
Sentinel-2 data for Austria of about 6m compared to a high-resolution regional DEM. The performance requirement of
Landsat-8 OLI for geometric terrain corrected accuracy is specified as 12m (Storey et al, 2014). Kääb et al. (2016) find
cross-track offsets of 20-30 m over glacier termini in the Swiss Alps when comparing Landsat-8 and Sentinel-2 scenes
acquired on September 8, 2015. Accordingly, uncertainties regarding the GPS points of the in situ measurements as
delineated in our sensitivity analysis (Table 4, Figure 11) can be considered relatively small compared to the those
related to the orthorectification of the satellite data. Comparisons between in situ point data and pixel values from the
satellite products must be interpreted keeping positional uncertainties in mind.
Decreasing the pixel resolution and averaging over multiple in situ measurement points can serve as an approach to
reduce the influence of geometric errors. However, any sort of averaging procedure must also be assessed in terms of
spatial representativeness of the point measurements for a greater area and, conversely, the down sampled satellite data
for small scale surface processes. What can be considered representative will always be a question of scale and
application. The glacier surface at the study site is locally very heterogenous and hence prone to representativeness
errors (Wu et al., 2019). We selected the location of the in situ profile lines so that they cover what we consider to be the
typical surface features and types of a given section of the ablation zone and argue that our 20 m long profile lines with
equidistant measurements at least every two meters capture any variations that are likely to influence the corresponding
pixel values of the satellite data. Naturally, the less overlap there is between the profile lines and any given satellite
pixel, the more likely it is that the in situ point data happen to capture something that differs strongly from what the
satellite sees.
The different surface types identified at Jamtalferner (Fig. 4) and their reflectance spectra are comparable to types of
surfaces identified in Switzerland at Morteratsch and Glacier de la Plaine Morte by Di Mauro et al. (2017) and Naegeli
et al. (2015), respectively, supporting the use of a classification scheme based on differentiating between a) clean and
dirty ice surfaces and b) the presence or absence of liquid water on the ice surface. Classifying the surface
characteristics into discrete types can help to ensure representativeness e.g. by quantifying how much of a given area
subsection relevant to the comparison with remote sensing data is comprised of which type and then sampling
accordingly. However, surface types are not always discrete in practice. Nicholson and Benn (2006) indicate that the
surface albedo of ice with scattered debris can be simulated in a modelling approach be linearly varying between clean
ice albedo values and values for debris, but this does not necessarily account for other types of surfaces and even the
clean ice albedo can vary considerably, especially if liquid water is present. Additionally, classification by type of any
kind cannot address the issue of temporal representativeness unless the temporal variability of different surface types is
first determined.
Profile 8 shows particularly poor agreement with the corresponding satellite data and may be an example where
temporal variability plays a role: The profile crosses a section of ice where the contrast between dark and bright areas is
comparatively strong. The profile line is roughly at a right angle to the flow direction of the glacier and "stripes" of
meltwater channels and/or dirt cross the line. The profile has a comparable number of individual spectra with
reflectance values above and below the profile mean, i.e. it is not a dark profile with a few bright outliers (compare e.g.
to P6 in Fig. 6) or vice versa (e.g. P3), but alternates along the profile line. Agreement with the remote sensing data is
decent for the darker spectra in P8 but the bright values are not captured. While we cannot rule out that the lack of
agreement between the field and remote sensing data is due to an unusually unfortunate/unrepresentative positioning of
the field measurement points in the satellite pixels, this may be an instance where the diurnal melt cycle and the
associated presence/absence of water on the surface exacerbates the contrast between the dark and bright sections of the
profile. In the bright sections, the porous weathering crust and cryconite hole structures appear to be drained of water,
while the depressions of the melt channels are noticeably wet. Cook et al. (2016) indicate the occurrence of "sudden
drainage events" in the weathering crust on a day-to-day time scale and a diurnal cycle of the hydrology of the
weathering crust driven by meteorological conditions (radiation, turbulent fluxes). The time of day of a satellite
overpass would determine which stage of this cycle the satellite sees and consequently the satellite data would not
capture this variability. In order to assess how much the time of day of the overpass could systematically affect the
representativeness of the satellite date for actual ground reflectance, it needs to be determined how significant and how
consistent the diurnal cycle is. To do this, the driving processes must be identified, keeping in mind that these may be
different for different types of glaciers and that different causes of short-term albedo change can overlap. E.g.: Azzoni et
al. (2016) point out that meltwater increases albedo around midday in a daily cycle, while rain causes increased albedo
for up to 4 days after the precipitation event. A seasonal cycle of albedo has been demonstrated in previous
observational studies and modelling efforts of broadband albedo, highlighting the importance of continuous
measurements (e.g. Hoinkes and Wendler, 1968; Nicholson and Benn, 2012; Möller and Möller, 2017).

**4.3. Relevance of small-scale variability**

The reflectance properties of ice are a central part of mass and energy balance modelling, usually in the form of a
glacier wide broadband albedo, or using one value for ice in the ablation zone and one for snow covered areas.
Resolving local albedo variations at a very small, sub-pixel scale is not required for regional or global studies, provided
the albedo parametrization captures the conditions on the ground adequately for the region of interest. In their important
2015 study, Naegeli et al. find that Sentinel-2 and Landsat-8 reflectance data are within the suggested accuracy
requirements for global climate modelling (±0.05, Henderson-Sellers and Wilson, 1983) over their study site, Glacier de
la Plaine Morte in Switzerland. In the same study, they report a 10% difference in modelled mass balance when a
spatially distributed albedo is used to force the model as opposed to a single, glacier wide albedo. Significantly larger
differences occur in parts of the glacier where water is present on the surface or the ice surface contains a lot of light-
absorbing impurities. While the glacier-wide impact of a spatially distributed albedo on model results may be relatively
small, this highlights that resolving local variability of reflectance properties and its causes is important for accurately
predicting the future evolution of individual glaciers, especially in cases where the firn covered area is gone or greatly
reduced and rapid melt is occurring. Only once the problem of different scales comparing point and spatially averaged
data is solved, the relationship between albedo variability and mass balance point and averaged data can be tackled to
calculate the effects on mass balance at glacier-wide or regional scale.
Aside from directly mass and energy balance related applications, reflectance data with high spatial and temporal
resolution is essential to improve understanding of micro-hydrological processes in the weathering crust and how these

may affect a possible larger scale darkening of increasingly snow free glaciers, e.g. by favoring or impeding the growth of ice algae, or the collection/washing out of cryoconite or other impurities. High resolution time series of spectral reflectance at representative locations in the ablation zone are needed to assess how changes in wetness and temperature, surface texture (cryoconite formation, roughness changes during the season), biotic productivity, deposition of sediment by melt water and rain affect reflectance properties on a small spatial scale, throughout the day and over the course of the ablation season. Establishing measurement efforts aimed at generating such time series on glaciers with existing mass balance monitoring networks would be highly desirable in order to better link small scale surface processes with mass and energy balance modelling.

**5. Conclusion & Outlook**

In comparing our in situ measurements with readily available L2A satellite products, we chose an "as simple as possible" approach to gain a general understanding of where sources of uncertainties are. We found that the difference between in situ and satellite data is not uniform across satellite bands, between Landsat and Sentinel, and to some extent between surface types. Reflectance variability on the ground is not fully represented in the satellite data, which raises questions as to how well surface processes at rapidly changing glaciers such as Jamtalferner can be resolved with satellite data.

The reflectance properties of ice, along with other feedback mechanisms such as changing topography and glacier geometry, significantly impact the rate of glacial retreat, contributing to the non-linear characteristics of glacier change and the high variability of defining parameters such as mass-balance or area change even among neighbouring glaciers subject to common climatic drivers (Charalampidis et al., 2018). Understanding these feedback mechanisms and associated processes is key to successfully predicting future glacier changes across spatial and temporal scales. Ice albedo will remain a significant source of uncertainty in modelling applications as long as the processes governing temporal and spatial variability are not fully understood.

Quantifying spatial and temporal variability of spectral reflectance and delineating the main causes of this variability for individual glaciers will improve modelling capabilities of glacier evolution and catchment hydrology. Satellite-derived reflectance products are a key component of tackling similar questions on the regional and global level. However, ground truth data from representative sites is essential in order to understand uncertainties associated with satellite albedo and surface reflectance products and potentially improve them for specific contexts.

Moving forward, an expansion of the monitoring network at Jamtalferner and, ideally, other glaciers, by continuous reflectance measurements in the ablation zone at a fixed location is needed, as well as "snap-shot" measurements of spectral, multi-angular reflectance at multiple strategic points in regular intervals. Combining analysis of spectral reflectance data from in situ and remote sensing sources with the wealth of contextual information available at Jamtalferner and other established monitoring sites has the potential to greatly improve our understanding of the complex interplay of surface changes, glacier dynamics, and mass- and energy balance.

**Author Contribution**

L. Felbauer and A. Fischer collected the in situ data. Subsequent data curation was carried out by L. Felbauer and L. Hartl. G. Schwaizer conceptualized the comparison of in situ and satellite derived data. L. Hartl developed the code for data analysis and visualizations, and wrote the manuscript with contributions from all co-authors.

**Competing interests**

The authors have no competing interests to declare.

**Data availability**

The spectral reflectance data can be downloaded at: https://doi.pangaea.de/10.1594/PANGAEA.915932
And interactively explored in a web-app at: http://spectralalbedo.mountainresearch.at/

**Acknowledgements**

We are very grateful to Gottlieb Lorenz and the entire team at the Jamtal Hütte for providing an excellent base for field work at Jamtalferner and invaluable support over the years. We sincerely thank M. Pelto and the second, anonymous reviewer for their helpful comments!

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
