# Peer review of "Small scale spatial variability of bare-ice reflectance at Jamtalferner, Austria"

_The Cryosphere, 2020_

## Referee Comment (RC1) · Mauri Pelto (Referee) · 12 Jun 2020

This study by Hartl et al (2020) compares a detailed field survey of albedo on Jamtalferner with synchronous remote sensing derived albedo from Sentinel and Landsat images. The methods for both approaches to albedo determination are well explained. The comparison of the field albedo and remote sensing derived albedo is the key output of this paper and is well illustrated in Figures 7-9. The study provides a richer data set for understanding how Landsat or Sentinel images could be used and is simply interesting. The primary comments below are seeking more context: 1) On the value of detailed spatial and temporal albedo observations. 2) For connections with energy balances. I am not suggesting additional data or figures be presented, but instead additional reference to other work and how the data here fits with these.

9: The first sentence reverses the cause and effect. "As Alpine glaciers recede, they are quickly becoming snow free in summer and, accordingly, spatial and temporal variations in ice albedo increasingly affect the melt regime. " Instead I suggest, "As alpine glacier become snow free in summer, recession occurs, and further spatial and temporal variations in ice albedo increasingly accentuate the melt regime."

16: Finishing the sentence with fluid is confusing since that could be a surface type, "Spectra can roughly be grouped into dry ice, wet ice, and dirt/rocks, although transitions between types are fluid." Maybe finish with, "although gradations between these groups occur". Replace "fluid" with gradations throughout.

24: Explain that firn cover is lost when persistent loss of snow cover in the accumulation zone exposes the firn (Fischer, 2011).

59: "…relatively recent times", be more specific.

74: Azzoni et al. (2016) also found a significant impact from rain water.

76: What is the basis for Brun et al (2015) stating importance of remote sensing in albedo assessment?

77: Resolution of Naegeli et al. (2015) aerial albedo observations?

96: Is it worth observing that for degree day modelling changing albedo with time would alter parameters in the model.

109: Given the illustrations in Figure 2 leverage these with terminus retreat from 1990-2017 and for the accumulation zone what is the mean AAR during this same period 1990-1999, 2000-2009 and 2010-2017?

117: "Along each profile line spectra are gathered at equal intervals, with 14 profile lines containing 11 spectra spaced at 2(?)m and 2 profiles containing 40 spectra gathered at a higher resolution of 0.5(?) m." 132: "Google Earth Engine"

161: "gradational" instead of "fluid"

196: Profile 8 seems to have the least agreement in Figure 9 between field are remote sensing data, why?

204: Figure 8 has excellent potential for the direct spatial correlation of the Sentinel albedo to the point measurements. I think showing all the profiles prevents being able to visualize the relationship. I suggest focusing on a few of the same profiles that were a focus of Figure 5 and provide a range of conditions ie. P 3, 5, 8, and 11. Anzoni et al (2016) noted a future goal of generating an albedo map. Is that feasible for the area of the glacier shown in Figure 1?

210: This is a key observation. What have other studies found in terms of the over/under-estimate transition?

226: The variation in energy balance as albedo/debris cover changes spatially and temporally was a focus of Nicholson and Benn (2006) provided a nice overview of this from Ghiacciaio del Belvedere. They observed for debris cover areas the dominant energy contribution varied from sensible heat to shortwave radiation due to decreased albedo and higher surface temperatures. They further found that for dry debris cover, sensible heat flux became negative as debris cover thickened, because of higher surface temperatures and that longwave radiation became negative even for thin debris cover.

231: How significant is the time of day variation in albedo? How consistent would this variation be from day to day? Moller and Moller (2017) provide one measure of this in an examination of spatiotemporal variations of albedo across Svalbard glaciers, recognizing this is a larger scale model albedo product. Nicholson and Benn (2012) examining Ngozumpa Glacier identify surface albedo variation across an area of varied debris cover, as well as the changing diffusivity through the melt season. The surface temperature variation of this glacier in the Himalaya would be much different than in the Alps, yet the continuous record compiled does provide context to the degree of variation and the potential importance of ongoing point measurments. They observe

the importance of distinguishing wet vs dry surfaces. Azzoni et al (2016) note the increased albedo due meltwater presence during the middle of the day to albedo, while rain led to increased albedo for several days.

249: Similarly, the question of how well the albedo variations need to be resolved to model or understand surface processes need to be acknowledged/discussed. One reason a relatively sparse ablation stake network can represent ablation during a melt season is that despite significant surface changes the spatial distribution of energy balance over time tends to balance. Your Figure 5 illustrates this that though albedo varies considerably along the Profile 3 and 11, and the profiles have been exposed ablation ice for some period, the ice surface is relatively even. Energy balance distribution across an ice surface in a small area responds to the variations in surface level, albedo and debris cover.

260: A significant source of uncertainty for what?

271: Need a reference from a different region to emphasize this point.

280: Did you sample spectra at any location over a period of time? If so, this helps relate the logistical challenge of temporal albedo monitoring.

Azzoni, R., Senese, A., Zerboni, A., Maugeri, M., Smiraglia, C. and Diolaiuti, G.: Estimating ice albedo from fine debris cover quantified by a semi-automatic method: the case study of Forni Glacier, Italian Alps, The Cryosphere, 10, 665- 679, 2016.

Moller, M. and Moller, R.: Modeling glacier-surface albedo across Svalbard for the 1979–2015 period: The HiRSvaC500-a data set. J. Adv. Model. Earth Syst., 9, 404–422, 2017.

Nicholson, L. and Benn, D.: Calculating ice melt beneath a debris layer using meteorological data. Journal of Glaciology 52(178): 463–470, 2006.

Nicholson, L. and Benn, D.: Properties of natural supraglacial debris in relation to modelling sub‐debris ice ablation. Earth Surf. Process. Landforms, 38: 490-501,

doi:10.1002/esp.3299, 2012.

---

## Referee Comment (RC2) · Anonymous Referee #2 · 18 Aug 2020

In this paper, the authors present a comparison between spectral reflectance measurements of bare ice carried out in the ablation zone of the Jamtalferner glacier, Austria with concurrent Sentinel-2 and Landsat-8 acquisitions. In a first step, the spatial variability of the manually acquired surface albedo across the ablation zone of the glacier is presented, highlighting large differences in reflective properties from dry clean ice to surfaces covered in mineral and organic debris. Secondly, the paper focusses on comparing the field measurements with atmospherically-corrected satellite reflectance products to investigate whether physical processes related to deglaciation are fully captured by optical Earth Observation sensors. Results show that the differences observed between the ground-based and satellite measurements are not uniform depending on the wavelength, the sensor or surface type. The authors conclude by suggesting that further in-situ monitoring efforts are needed to be able to use satellite-derived reflectance for glacier change monitoring.

General assessment

The comparison of in-situ surface reflectance measurements with satellite-derived products is of great interest for anyone involved in space-borne observations of glaciers and more generally glacier surface processes monitoring, and in that sense, the work here is timely and most welcome. I particularly commend the use of openly accessible world-wide available satellite data rather than higher-resolution commercial data, making the applications available to a wider audience. The article is overall well written, apart from a couple of minor approximations (see detailed comments). However, the manuscript presents two major shortcomings that leave the reader missing significant information (see General comments paragraph below).

In summary, this article would have merit for publication in The Cryosphere if the major points referred to below are addressed. Currently, the Methods and Discussion sections are insufficient.

General comments

The first deficiency mentioned in the paragraph above concerns the presentation of the Methods. The ground measurements of spectral reflectance presented in Section 2.2 (7 lines) are largely insufficient for a piece of work dedicated to comparing ground measurements to satellite products. Indeed, the section barely skims over the way measurements were collected and crucial information is lacking to clearly understand the comparisons made.

1. When were the measurements collected? No date or time of measurements is provided in the section describing ground measurements. The reader has to wait until Section 2.3 to understand that the measurements were acquired on 4th September 2019. Over what time period (start and end of acquisitions) was the data acquired?

This is of significant importance for the comparison of the data, e.g. did the surface have time to change between the satellite overpass and the ground measurements?

2. There is no description of the environmental conditions during the acquisition, e.g cloud cover. Even a small amount of cloud cover, such as the presence of rapidly changing cirrus can introduce uncertainties of several percent in the measured reflectance.

3. The method for measuring the distance between the points on the profile is indicated, but how were the measurements geo-located in the field? Were there any GPS points acquired (especially as the authors refer to "GPS profile" in figure 1), with what uncertainty? The uncertainty in the positioning of the ground spectra may impact your point-to-pixel comparisons (to be addressed in the Discussion also).

4. The measurement protocol is not described sufficiently, leaving the reader with a number of interrogations: how were the measurements carried out: was the ASD fibre optic handheld or placed on a device to reduce operator interference (Fig 3 in Wright et al. 2014, Kimes et al. 1983)? Did the authors use an optical lens on the fibre optic (if so, what field-of-view)? What height was the collector from the surface / spectral panel when performing the measurements? A description of how the measurements were performed is desired, or at the least, if the authors were following an existing protocol, a reference to the article is expected.

5. The description of the processing of the raw ASD is missing. There are numerous steps to be carried out during the processing of data, including the application of instrument or spectral calibration files. In the current state, the description of the processing is too vague.

6. The authors are not clear about the physical quantities measured. The title reads "Small scale variability of bare-ice albedo at Jamtalferner, Austria", and the author summarise the body of work on broadband and spectral albedo. However, in the methods, the field acquisitions are referred to as spectral reflectance and the (limited) description

of the measurement protocol leads the author to believe that the authors are recording hemispherical–conical reflectance. The ground measurements are then compared to surface reflectance products derived from Sentinel-2 and Landsat-8. Particular care should be observed when describing remotely sensed quantities and I recommend that the authors verify inconsistencies throughout the paper. Very useful references in that sense are Schaepman-Strub et al., 2004, 2006 (besides an important corpus on the subject).

The second shortfall mentioned in the overall remarks concerns the Discussion, that does not do justice to the paper. Indeed, in its current state, the section repeats the introduction and doesn't address the rich results obtained by the authors. The key points presented in the results are barely brushed past and the discussion on the limitations of the methods employed and possible explanations for the results obtained are missing. The paragraph starting P8, L247 would deserve (consequential) expanding in regard to the results obtained. By restructuring the Discussion section, significant value could be brought to this otherwise valuable contribution to the observation of glacier ablation zones based on optical Remote Sensing.

Specific comments

- P1, L14: in the Optical Remote Sensing community, ground reflectance is commonly referred to as Bottom-Of-Atmosphere (BOA) reflectance. I am not suggesting to replace the term, but maybe add a mention to BOA.

- P1, L27: "The magnitude and [...] local production rates." > Although you go into further details later in the introduction, citations are missing here.

- P4, L106: Figure 2 and 3 seem irrelevant in the context of this paper that focusses on the comparison of ground and satellite acquisitions of reflectance and not the evolution of the surface properties over time. I suggest their removal, as they cloud the overall message. Rather, the satellite images (used in the study), of the glacier tongue with the profiles overlaid would be a nice addition to the paper.

- Section 2.3: Table 3 would benefit being completed with additional information on the Sentinel-2 and Landsat-8 acquisitions, such as acquisition time or the angular information (solar and viewing angles). A column with the corresponding ground measurement information would be a plus.

- P5, L126: The acquisition time of Sentinel-2 is not specified: yet this information is important to investigate the differences between the measurements from both sensors.

- P5, L139: Did the authors consider integrating the spectral measurements using the (available at least for Sentinel-2) spectral response of each band? Do the authors think that the difference with the average would be negligible or not?

- P6, L175: This is an interesting find. Have the authors considered the difference in viewing/solar geometries between the two acquisitions? The strong anisotropy of the ice could partly explain the differences (see the previous comment). Basic simulations of ice reflectance (using e.g. Malinka et al. 2016) could help investigate this point. To be clear, this is not expected from the authors, but a point that could be worth thinking about for future studies. Another factor that could influence the differences observed could be the different atmospheric corrections schemes used (a reference in the Discussion would be of value).

- P6, L183: This suggests that for surfaces with strong sub-pixel variability the resolution of the images is essential for an accurate description of the surface. The representativeness of field sampling when comparing in situ measurements to satellite images is of particular interest in the snow and ice community. Did the authors consider investigating the sensitivity to resolution by degrading the 10m bands to 30 then 60 meters?

- P7, L200: Very interesting find, which links to the question of the representativeness of the in-situ sampling. It would be nice to see this point further discussed in the Discussion section.

- P7, L206: Again, this key result deserves some discussion.

- P8, L222-226: the observation is repeated from the introduction.

- P8, L228: This paragraph should be placed in the context of the results of this study and is overall too vague.

- P8, L234: Again, the paragraph reads like an introduction and doesn't have a place in the discussion.

- P8, L244: Some lines of reflection in the context of the authors' study, such as discussing the anisotropy of ice in line with the differences in overpass geometries would be most welcome here.

- Figure 4: is the highlighting of the maximum and minimum spectra necessary? A single emphasised black spectrum of the mean and the others in light grey could be clearer (if the authors agree).

- Figure 6: in the printed manuscript, the tape measure is unreadable in the photos. Adding a small simple scale bar int the pictures would help grasp the scale of the images. This is an interesting figure showing the important variability of reflectance across the glacier.

- Figure 7: the caption is unclear and the reader has to read Section 3.2 several times to understand the figure. The term "ground measurements" for satellite images (P20, L419) is confusing. I would suggest revising the caption to clearly state what the blue and orange bars represent.

- Table 1: why are the PROMICE network measurements not referenced (Fausto and van As 2019)? They have been used for satellite calibration also.

Technical corrections

- P1, L12: exits > exist.

- P1, L16: at dark spectra > for dark spectra

- P1, L 25: "so that darker bare ice is exposed" > I suggest specifying "in Summer" to be more precise.

- P2, L33: "gap of knowledge" > "knowledge gap"

- P2, L39: "comparatively high resolution" > Comparatively to what? Please be more specific. Sentinel-2 and Landsat-8 could be referred to as "medium resolution sensors".

- P2, L59: "in relatively recent times" > Please be more specific.

- P3, L86: "different kinds of remote sensing" > this phrasing is a little vague, could you clarify?

- P4, L122: "specdal" > "spectral"

- Figure 9: please specify the wavelength of band 3.

- Table 2: is lacking the first column header

- Table 1, 2 and 3: I am guessing that the authors will format the tables correctly in the next iteration? They are currently unpleasant to read.

References

Fausto, R.S. and van As, D., (2019). Programme for monitoring of the Greenland ice sheet (PROMICE): Automatic weather station data. Version: v03, Dataset published via Geological Survey of Denmark and Greenland. DOI: https://doi.org/10.22008/promice/data/aws

Kimes, D. S., J. A. Kirchner, and W. Wayne Newcomb. "Spectral radiance errors in remote sensing ground studies due to nearby objects." Applied optics 22.1 (1983): 8-10.

Malinka, Aleksey, et al. "Reflective properties of white sea ice and snow." Cryosphere 10.6 (2016).

Schaepman-Strub, G., et al. "About the importance of the definition of reflectance quantities-results of case studies." Proceedings of the XXth ISPRS Congress. 2004.

Schaepman-Strub, Gabriela, et al. "Reflectance quantities in optical remote sensing—Definitions and case studies." Remote sensing of environment 103.1 (2006): 27-42.

Wright, Patrick, et al. "Comparing MODIS daily snow albedo to spectral albedo field measurements in Central Greenland." Remote Sensing of Environment 140 (2014): 118-129.

---

## Author Comment (AC1) · 20 Aug 2020

Author responses are below the respective reviewer comments and separated by —- throughout the document.

Reviewer comment:

This study by Hartl et al (2020) compares a detailed field survey of albedo on Jamtalferner with synchronous remote sensing derived albedo from Sentinel and Landsat images. The methods for both approaches to albedo determination are well explained. The comparison of the field albedo and remote sensing derived albedo is the key output of this paper and is well illustrated in Figures 7-9. The study provides a richer data set for understanding how Landsat or Sentinel images could be used and is simply

interesting. The primary comments below are seeking more context: 1) On the value of detailed spatial and temporal albedo observations. 2) For connections with energy balances. I am not suggesting additional data or figures be presented, but instead additional reference to other work and how the data here fits with these.

———

Author response:

Thank you for your comments and suggestions. We propose to address the main issues raised above in a revised version of the manuscript primarily by expanding the discussion section to include more depth on points 1 and 2 (value of detailed spatial and temporal observations, connection with energy balance), contextualizing our results in the existing body of work more clearly and exploring in more detail the potential for future work on how small scale reflectance variability may affect energy balance. Additionally, we will rework parts of the introduction and add more comments on existing literature in order to better set the stage for approaching the issues related to points 1 and 2.

We address the specific comments below. We are happy to follow all suggestions within the constraints of the data available to us.

———

9: The first sentence reverses the cause and effect. "As Alpine glaciers recede, they are quickly becoming snow free in summer and, accordingly, spatial and temporal variations in ice albedo increasingly affect the melt regime. " Instead I suggest, "As alpine glacier become snow free in summer, recession occurs, and further spatial and temporal variations in ice albedo increasingly accentuate the melt regime."

———

Will rephrase as suggested.
‾‾

16: Finishing the sentence with fluid is confusing since that could be a surface type, "Spectra can roughly be grouped into dry ice, wet ice, and dirt/rocks, although transitions between types are fluid." Maybe finish with, "although gradations between these groups occur". Replace "fluid" with gradations throughout.

‾‾

Will rephrase as suggested and check manuscript for use of "fluid".

‾‾

24: Explain that firn cover is lost when persistent loss of snow cover in the accumulation zone exposes the firn (Fischer, 2011).

‾‾

Will add note on this and citation.

‾‾

59: ". . .relatively recent times", be more specific.

‾‾

Propose to replace with "...throughout the last decade".

‾‾

74: Azzoni et al. (2016) also found a significant impact from rain water.

‾‾

Will change "as well as significant effects of melt water" to: "as well as significant effects of melt and rain water"

‾‾

76: What is the basis for Brun et al (2015) stating importance of remote sensing in albedo assessment?

⎯⎯

Brun et al (2015) point out that satellite products are critical for glaciological studies in data sparse regions such as the Himalayas, where their study sites are, as in situ data are often not available and glaciers may not be easily accessible. In their study they reconstruct annual mass balance from MODIS albedo data for two glaciers, validating this with in situ data. They suggest that this method can be applied to certain other glaciers in the HKH region from which no in situ mass balance or albedo data is available. This highlights that 1) remote sensing is often the only way of getting albedo data and 2) other important glaciological work can be carried out using remote sensing derived albedo data.

⎯⎯

77: Resolution of Naegeli et al. (2015) aerial albedo observations?

⎯⎯

From Naegeli et al. 2015: "The flying altitude of 4000 m above ground level (a.g.l.) in combination with an instantaneous field of view (FOV) of 0.0025° resulted in a surface projected pixel resolution of ∼ 2 m."

⎯⎯

96: Is it worth observing that for degree day modelling changing albedo with time would alter parameters in the model.

⎯⎯

Propose to add: "In addition, delineating temporal variability of albedo is relevant to degree day modelling, as a changing albedo would alter parameters in the model."

—-

117: "Along each profile line spectra are gathered at equal intervals, with 14 profile lines containing 11 spectra spaced at 2(?)m and 2 profiles containing 40 spectra gathered at a higher resolution of 0.5(?) m." // 132: "Google Earth Engine" // 161: "gradational" instead of "fluid"

Will change as suggested.

—-

196: Profile 8 seems to have the least agreement in Figure 9 between field are remote sensing data, why?

—-

We have attached photos of the profile at the end of this document to provide visual context. Profile 8 crosses a section of ice where the contrast between dark and bright areas is comparatively strong. The profile is roughly at a right angle to the flow direction and there are "stripes" of meltwater channels and/or dirt that cross the profile. The profile has a comparable number of individual spectra with reflectance values above and below the profile mean, i.e. it is not a dark profile with a few bright outliers (compare e.g. to P6 in Fig 8) or vice versa (e.g. P3), but alternates along the profile line. Agreement with the remote sensing data is decent for the darker spectra in P8 but the bright values are not captured.

While we cannot rule out that the lack of agreement between the field and remote sensing data is due to an unusually unfortunate/unrepresentative positioning of the field measurement points in the satellite pixels, this may be an instance where the diurnal melt cycle and the associated presence/absence of water on the surface exacerbates the contrast between the dark and bright sections of the profile. In the bright sections, the porous weathering crust and cryconite hole structures appear to be drained of water, while the depressions of the melt channels are noticeably wet. Cook et al.

2015 (https://doi.org/10.1002/hyp.10602) indicate the occurrence of "sudden drainage events" in the weathering crust on a day-to-day time scale and a diurnal cycle of the hydrology of the weathering crust driven by meteorological conditions (radiation, turbulent fluxes). The time of day of a satellite overpass would determine which stage of this cycle the satellite "sees" and consequently the satellite data would not capture this variability. A more definitive explanation would require further study and dedicated field experiments designed specifically to explore this aspect of reflectance variability – we hope to do this in the future.

We propose adding a version of the above commentary to the manuscript to explain the particularities of profile 8.

—-

204: Figure 8 has excellent potential for the direct spatial correlation of the Sentinel albedo to the point measurements. I think showing all the profiles prevents being able to visualize the relationship. I suggest focusing on a few of the same profiles that were a focus of Figure 5 and provide a range of conditions ie. P 3, 5, 8, and 11. Anzoni et al (2016) noted a future goal of generating an albedo map. Is that feasible for the area of the glacier shown in Figure 1?

—-

We are happy to change Figure 8 so that it shows only selected profiles, as suggested.

Based on high resolution, close range digital images of the ice surface at Forni glacier, Anzoni et al (2016) develop a relationship between the area ratio of ice covered by fine debris and clean ice (d) such that albedo can be derived for a given area if d is known. To apply this method to the area of the glacier in Figure 1 would require an estimate of d, for which we would need close range imagery of the ice surface for the entire area. We could perhaps apply the method of Anzoni et al. to the photos we took of the ice surface at our sampling sites, but this would still result in albedo values only at our

sampling sites without addressing the question of how representative these locations are for the rest of the glacier area and how albedo might be interpolated between them. We hope that our work may eventually contribute to methods for producing high resolution albedo maps, but do not think making such a map is feasible with our current dataset and the method described by Anzoni et al.

—-

210: This is a key observation. What have other studies found in terms of the over/under-estimate transition?

—-

We have not been able to find many other studies with explicit information on this issue specifically over glacier ice surfaces. Hendricks et al. (2004) state for measurements at Hintereisferner: "Except for ice, the glacier reflectances derived from the satellite image are large underestimations when comparing them to the spectrometer measurements. A maximum underestimation of 139 % was found for firn in band 4. New snow, with the highest reflectance of 86 % is predicted most accurately within a confidence interval of 15 - 18 %. The reflectance of ice seems to be highly variable with both under - and overestimations of up to 76 % and 31 % respectively." This refers to Landsat ETM+ imagery acquired about 2 weeks before the corresponding field measurements. We propose to cite this in the revised manuscript with additional discussion of the effects of changing atmospheric conditions, which make direct comparisons of ground measurements and satellite data challenging if measurements and satellite overpasses do not coincide closely, especially at a comparatively small spatial scale.

—-

226: The variation in energy balance as albedo/debris cover changes spatially and temporally was a focus of Nicholson and Benn (2006) provided a nice overview of this from Ghiacciaio del Belvedere. They observed for debris cover areas the dominant
energy contribution varied from sensible heat to shortwave radiation due to decreased albedo and higher surface temperatures. They further found that for dry debris cover, sensible heat flux became negative as debris cover thickened, because of higher surface temperatures and that longwave radiation became negative even for thin debris cover.

—-

We will add comments discussing the results of Nicholson and Benn (2006) in this section, with notes on how they relate to our findings.

—-

231: How significant is the time of day variation in albedo? How consistent would this variation be from day to day? Moller and Moller (2017) provide one measure of this in an examination of spatiotemporal variations of albedo across Svalbard glaciers, recognizing this is a larger scale model albedo product. Nicholson and Benn (2012) examining Ngozumpa Glacier identify surface albedo variation across an area of varied debris cover, as well as the changing diffusivity through the melt season. The surface temperature variation of this glacier in the Himalaya would be much different than in the Alps, yet the continuous record compiled does provide context to the degree of variation and the potential importance of ongoing point measurments. They observe the importance of distinguishing wet vs dry surfaces. Azzoni et al (2016) note the increased albedo due meltwater presence during the middle of the day to albedo, while rain led to increased albedo for several days.

—-

We will expand this section with a discussion of the above points. We will discuss our thoughts on the potential magnitude of day to day variations, the regularity of the diurnal cycle, and meteorological factors affecting the cycle in the context of our study as well as the relevant literature (citations by the reviewer, as above, and others, e.g.

Cook et al., 2015).

In order to quantitatively answer the questions posed at the beginning of this comment, we very much hope to expand our data collection at Jamtalferner and install instrumentation that would allow continuous point measurements.

—-

249: Similarly, the question of how well the albedo variations need to be resolved to model or understand surface processes need to be acknowledged/discussed. One reason a relatively sparse ablation stake network can represent ablation during a melt season is that despite significant surface changes the spatial distribution of energy balance over time tends to balance. Your Figure 5 illustrates this that though albedo varies considerably along the Profile 3 and 11, and the profiles have been exposed ablation ice for some period, the ice surface is relatively even. Energy balance distribution across an ice surface in a small area responds to the variations in surface level, albedo and debris cover.

—-

"Similarly, the question of how well the albedo variations need to be resolved to model or understand surface processes need to be acknowledged/discussed." - This is a valid and interesting point of discussion. We suggest that the answer to this question depends on the processes one is trying to understand and the scale at which they occur. In the context of glacier wide ablation monitoring via the direct glaciological method, resolving sub daily and sub meter variations is perhaps not exactly a very pressing need, but, in our opinion, still interesting. The area of the glacier shown in Figure 1 contains 9 ablation stakes, which we maintain as part of our mass balance monitoring program at Jamtalferner. We observe significant differences in the amount of melt that occurs at these stakes. Aspect, shading, and slope angle of course play a strong role in this, as does locally increasing debris cover. We hypothesize that darkening due to water on the glacier surface is more of a factor at some of our stakes than at others, depending
e.g. on their position in relation to seasonally shifting meltwater channels. We would like to eventually achieve a clearer separation of the influence of these factors (especially the influence of water), their relative magnitudes, and possible changes over time. We think that small scale reflectance monitoring can contribute valuable insights in this context.

Additionally, data with high spatial and temporal resolution seems essential to improve understanding of micro-hydrological processes in the weathering crust and how these may affect a possible larger scale darkening of increasingly snow free glaciers, e.g. by favoring or impeding the growth of ice algae, or the collection/washing out of cryoconite.

—-

260: A significant source of uncertainty for what?

—-

Surface reflectance and parameters that might be derived from it are key variables in glaciological modelling and uncertainty therein accordingly contributes to overall model uncertainty. This has implications for applications such as modelling runoff and catchment hydrology. We will clarify the phrasing in a revised version of the manuscript.

—-

271: Need a reference from a different region to emphasize this point.

—-

Propose to rephrase this sentence slightly to focus on changing debris cover as a source of uncertainty for future glacier change and to additionally cite Bolch et al., 2012, for the Himalayan region and Stokes et al., 2017, for the Caucasus.

—-

280: Did you sample spectra at any location over a period of time? If so, this helps

relate the logistical challenge of temporal albedo monitoring.

—-

We have not had opportunity to do that but hope we will in the future.

—-

References reviewer:

Azzoni, R., Senese, A., Zerboni, A., Maugeri, M., Smiraglia, C. and Diolaiuti, G.: Estimating ice albedo from fine debris cover quantified by a semi-automatic method: the case study of Forni Glacier, Italian Alps, The Cryosphere, 10, 665- 679, 2016.

Moller, M. and Moller, R.: Modeling glacier-surface albedo across Svalbard for the 1979–2015 period: The HiRSvaC500-a data set. J. Adv. Model. Earth Syst., 9, 404–422, 2017.

Nicholson, L. and Benn, D.: Calculating ice melt beneath a debris layer using meteorological data. Journal of Glaciology 52(178): 463–470, 2006.

Nicholson, L. and Benn, D.: Properties of natural supraglacial debris in relation to modelling subâ ̆A ̈Ÿ Rdebris ice ablation. Earth Surf. Process. Landforms, 38: 490-501, doi:10.1002/esp.3299, 2012.

References authors:

Bolch, T., Kulkarni, A., Kääb, A., Huggel, C., Paul, F., Cogley, J. G., ... & Bajracharya, S. (2012). The state and fate of Himalayan glaciers. Science, 336(6079), 310-314.

Cook, J. M., Hodson, A. J., & Irvineâ ̆A ̌RFynn, T. D. (2016). Supraglacial weathering crust dynamics inferred from cryoconite hole hydrology. Hydrological Processes, 30(3), 433-446.

Hendriks, J., and Petri. (2004). Estimation of reflectance from a glacier surface by comparing spectrometer measurements with satellite-derived reflectances. Zeitschrift
für Gletscherkunde und Glazialgeologie. 38, no. 2. 139-154.

Stokes, C. R., Popovnin, V., Aleynikov, A., Gurney, S. D., & Shahgedanova, M. (2007). Recent glacier retreat in the Caucasus Mountains, Russia, and associated increase in supraglacial debris cover and supra-/proglacial lake development. Annals of Glaciology, 46, 195-203.

[Figure]

[Figure]

**Fig. 1.** Profile 8, looking east.

[Figure]

**Fig. 2.** Profile 8, looking west.

---

## Author Comment (AC2) · 20 Aug 2020

We have copied the reviewer comments into this document and respond to them individually below. Author responses are below the respective reviewer comments and separated by —- throughout the document.

Reviewer comment:

In this paper, the authors present a comparison between spectral reflectance measurements of bare ice carried out in the ablation zone of the Jamtalferner glacier, Austria with concurrent Sentinel-2 and Landsat-8 acquisitions. In a first step, the spatial variability of the manually acquired surface albedo across the ablation zone of the glacier is presented, highlighting large differences in reflective properties from dry clean ice

to surfaces covered in mineral and organic debris. Secondly, the paper focusses on comparing the field measurements with atmospherically-corrected satellite reflectance products to investigate whether physical processes related to deglaciation are fully captured by optical Earth Observation sensors. Results show that the differences observed between the ground-based and satellite measurements are not uniform depending on the wavelength, the sensor or surface type. The authors conclude by suggesting that further in-situ monitoring efforts are needed to be able to use satellite-derived reflectance for glacier change monitoring.

General assessment

The comparison of in-situ surface reflectance measurements with satellite-derived products is of great interest for anyone involved in space-borne observations of glaciers and more generally glacier surface processes monitoring, and in that sense, the work here is timely and most welcome. I particularly commend the use of openly accessible world-wide available satellite data rather than higher-resolution commercial data, making the applications available to a wider audience. The article is overall well written, apart from a couple of minor approximations (see detailed comments). However, the manuscript presents two major shortcomings that leave the reader missing significant information (see General comments paragraph below).

In summary, this article would have merit for publication in The Cryosphere if the major points referred to below are addressed. Currently, the Methods and Discussion sections are insufficient.

—-

Author response:

We thank the reviewer for their time and the detailed and constructive commentary. The points of criticism are valid and we will address them (if given the opportunity) in a revised version of the manuscript, following the suggestions by both reviewers.

[Figure]

To remedy the main shortcomings of the Methods and Discussion sections as specified in this review, we propose to:

1) significantly expand the Methods section, particularly where it concerns the measurement protocol for the in situ data collection and the atmospheric conditions during data acquisition.

2) restructure and rewrite the discussion section to address the specific issues pointed out by the reviewer in the comments below. We accept that the current version is too vague and contains some parts that are better suited for the Introduction, while lacking detail in other areas. Both reviewers have made comments on specific issues that need to be expanded upon in the discussion and we agree that doing so will improve the manuscript.

We are happy to follow all suggestions made by the reviewer within the constraints of the data available to us.

—-

General comments

The first deficiency mentioned in the paragraph above concerns the presentation of the Methods. The ground measurements of spectral reflectance presented in Section 2.2 (7 lines) are largely insufficient for a piece of work dedicated to comparing ground measurements to satellite products. Indeed, the section barely skims over the way measurements were collected and crucial information is lacking to clearly understand the comparisons made.

—-

See response to specific comments below.

—-

1. When were the measurements collected? No date or time of measurements is

provided in the section describing ground measurements. The reader has to wait until Section 2.3 to understand that the measurements were acquired on 4th September 2019. Over what time period (start and end of acquisitions) was the data acquired? This is of significant importance for the comparison of the data, e.g. did the surface have time to change between the satellite overpass and the ground measurements?

—-

Ground measurements were taken on 4th September 2019, between approximately 10 am and 3 pm local time. The Sentinel overpass occurred at 10:20 GMT on Sept. 4. The Landsat overpass occurred at 10:10 GMT on Sept. 3. We will specify exact times for the first and last measured profiles in the revised manuscript and begin the section describing the ground measurements by stating the date and time period of the data collection. We propose adding a brief overview of how the surface may have changed over the course of these two days in this section, with more detailed considerations on the significance of possible changes for our analysis in the discussion section.

—-

2. There is no description of the environmental conditions during the acquisition, e.g cloud cover. Even a small amount of cloud cover, such as the presence of rapidly changing cirrus can introduce uncertainties of several percent in the measured reflectance.

—-

We will add a description of the environmental conditions during data acquisition in the methods section and will add commentary on possible uncertainties introduced by changing atmospheric conditions during the time period of the ground measurements in the discussion. The study site is free of cloud cover in both satellite images. Description of conditions during the ground measurements will be based on notes made by the field team and data from a nearby weather station.

—-

3. The method for measuring the distance between the points on the profile is indicated, but how were the measurements geo-located in the field? Were there any GPS points acquired (especially as the authors refer to "GPS profile" in figure 1), with what uncertainty? The uncertainty in the positioning of the ground spectra may impact your point-to-pixel comparisons (to be addressed in the Discussion also).

—-

GPS points were taken at the start and end point of each profile line, using a standard handheld GPS device. The horizontal accuracy for these devices is typically in the range of 3-5m. We will specify this further in the revised manuscript (considerations on terrain dependent accuracy and exact GPS model). We propose to add a quantitative estimation of the uncertainty in the point-to-pixel comparisons due to the GPS accuracy in the discussion.

—-

4. The measurement protocol is not described sufficiently, leaving the reader with a number of interrogations: how were the measurements carried out: was the ASD fibre optic handheld or placed on a device to reduce operator interference (Fig 3 in Wright et al. 2014, Kimes et al. 1983)? Did the authors use an optical lens on the fibre optic (if so, what field-of-view)? What height was the collector from the surface / spectral panel when performing the measurements? A description of how the measurements were performed is desired, or at the least, if the authors were following an existing protocol, a reference to the article is expected.

—-

The fibre optic was handheld and used without an optical lens, at a distance of 30cm above the ground. We based our usage of the ASD device on the descriptions in Naegeli et al. (2017) and Di Mauro et al. (2017), who carried out comparable measurements on glacier surfaces. We will reference these studies in this section and add specifics on where our protocol differs from theirs in the revised manuscript.

—-

5. The description of the processing of the raw ASD is missing. There are numerous steps to be carried out during the processing of data, including the application of instrument or spectral calibration files. In the current state, the description of the processing is too vague.

—-

We will add a step by step description of the data processing in the revised manuscript. We used a feature of our instrument that saves the white reference measurement to the RAM of the instrument. When this option is enabled, subsequent reflectance measurements are calculated with respect to the reference and the result of this calculation is saved to the output data file, such that there is no separate file for the reference.

—-

6. The authors are not clear about the physical quantities measured. The title reads "Small scale variability of bare-ice albedo at Jamtalferner, Austria", and the author summarise the body of work on broadband and spectral albedo. However, in the methods, the field acquisitions are referred to as spectral reflectance and the (limited) description of the measurement protocol leads the author to believe that the authors are recording hemispherical–conical reflectance. The ground measurements are then compared to surface reflectance products derived from Sentinel-2 and Landsat-8. Particular care should be observed when describing remotely sensed quantities and I recommend that the authors verify inconsistencies throughout the paper. Very useful references in that sense are Schaepman-Strub et al., 2004, 2006 (besides an important corpus on the subject).

—-

The reviewer's assessment here is correct. We will make sure to remove these inconsistencies and clearly define the measured quantities. Thank you for pointing out the publications by Schaepman-Strub et al., these are indeed very helpful.

—-

The second shortfall mentioned in the overall remarks concerns the Discussion, that does not do justice to the paper. Indeed, in its current state, the section repeats the introduction and doesn't address the rich results obtained by the authors. The key points presented in the results are barely brushed past and the discussion on the limitations of the methods employed and possible explanations for the results obtained are missing. The paragraph starting P8, L247 would deserve (consequential) expanding in regard to the results obtained. By restructuring the Discussion section, significant value could be brought to this otherwise valuable contribution to the observation of glacier ablation zones based on optical Remote Sensing.

—-

We accept the reviewer's criticisms of the discussion and will follow suggestions on how to improve it as stated previously. This will include a significantly expanded discussion of the content of the paragraph specified above.

Specific comments

- P1, L14: in the Optical Remote Sensing community, ground reflectance is commonly referred to as Bottom-Of-Atmosphere (BOA) reflectance. I am not suggesting to replace the term, but maybe add a mention to BOA.

—-

We will add a note on this in the revised manuscript.

—-

- P1, L27: "The magnitude and [. . .] local production rates." > Although you go into

further details later in the introduction, citations are missing here.

—-

We will rewrite/restructure this so that the citations are in the right place and more concisely match the information in the text.

—-

- P4, L106: Figure 2 and 3 seem irrelevant in the context of this paper that focusses on the comparison of ground and satellite acquisitions of reflectance and not the evolution of the surface properties over time. I suggest their removal, as they cloud the overall message. Rather, the satellite images (used in the study), of the glacier tongue with the profiles overlaid would be a nice addition to the paper.

—-

We are happy to follow these suggestions and replace the current figures 2 and 3 with new figures showing the satellite images and profiles.

—-

- Section 2.3: Table 3 would benefit being completed with additional information on the Sentinel-2 and Landsat-8 acquisitions, such as acquisition time or the angular information (solar and viewing angles). A column with the corresponding ground measurement information would be a plus.

—-

We will add additional columns as suggested.

—-

- P5, L126: The acquisition time of Sentinel-2 is not specified: yet this information is important to investigate the differences between the measurements from both sensors.

—-

We will specify this in the text and add it to table 3 as suggested in the previous comment.

—-

- P5, L139: Did the authors consider integrating the spectral measurements using the (available at least for Sentinel-2) spectral response of each band? Do the authors think that the difference with the average would be negligible or not?

—-

We expect that the differences would vary, potentially strongly, depending on the band and will assess whether it is feasible to include this as an additional part of the analysis.

—-

- P6, L175: This is an interesting find. Have the authors considered the difference in viewing/solar geometries between the two acquisitions? The strong anisotropy of the ice could partly explain the differences (see the previous comment). Basic simulations of ice reflectance (using e.g. Malinka et al. 2016) could help investigate this point. To be clear, this is not expected from the authors, but a point that could be worth thinking about for future studies. Another factor that could influence the differences observed could be the different atmospheric corrections schemes used (a reference in the Discussion would be of value).

—-

We agree that the differences in the geometries/anisotropy are important considerations in this context and should be discussed in greater detail in relation to our results. We will look into the work of Malinka et al. in this context but agree that the complexities of simulating reflectance are such that they are probably better addressed separately in future work. We will also make sure to add some comments on the issue of atmospheric correction schemes in the discussion.

—-

- P6, L183: This suggests that for surfaces with strong sub-pixel variability the resolution of the images is essential for an accurate description of the surface. The representativeness of field sampling when comparing in situ measurements to satellite images is of particular interest in the snow and ice community. Did the authors consider investigating the sensitivity to resolution by degrading the 10m bands to 30 then 60 meters?

—-

We think this would be an interesting addition to the analysis and will included a sensitivity analysis in the revised manuscript.

—-

- P7, L200: Very interesting find, which links to the question of the representativeness of the in-situ sampling. It would be nice to see this point further discussed in the Discussion section.

—-

We agree that representativeness of the in situ sampling is an important issue and will discuss this in more detail in the revised manuscript.

—-

- P7, L206: Again, this key result deserves some discussion.

—-

Will make sure to give this appropriate room in the discussion.

—-

- P8, L222-226: the observation is repeated from the introduction.

—-

Will remove/rephrase.

—-

- P8, L228: This paragraph should be placed in the context of the results of this study and is overall too vague.

—-

We will rephrase this to contextualize it better with our results and specify our thoughts on the importance of choosing appropriate spatial and temporal resolution for measurements depending on the processes to be studied in a more detailed manner.

—-

- P8, L234: Again, the paragraph reads like an introduction and doesn't have a place in the discussion.

—-

Noted & agreed.

—-

- P8, L244: Some lines of reflection in the context of the authors' study, such as discussing the anisotropy of ice in line with the differences in overpass geometries would be most welcome here.

—-

We will add more context and discussion of the effects of anisotropy, overpass geometries, and atmospheric conditions as they relate specifically to our study.

—-

- Figure 4: is the highlighting of the maximum and minimum spectra necessary? A

single emphasised black spectrum of the mean and the others in light grey could be clearer (if the authors agree).

—-

Will reconsider the coloring choices and assess how to make the figure clearer.

—-

- Figure 6: in the printed manuscript, the tape measure is unreadable in the photos. Adding a small simple scale bar int the pictures would help grasp the scale of the images. This is an interesting figure showing the important variability of reflectance across the glacier.

—-

We will edit the pictures in the figure to add scale bars.

—-

- Figure 7: the caption is unclear and the reader has to read Section 3.2 several times to understand the figure. The term "ground measurements" for satellite images (P20, L419) is confusing. I would suggest revising the caption to clearly state what the blue and orange bars represent.

—-

We will rephrase the figure caption to improve clarity.

—-

- Table 1: why are the PROMICE network measurements not referenced (Fausto and van As 2019)? They have been used for satellite calibration also.

—-

This was an unfortunate oversight on our side. We will add a reference to PROMICE

to the revised table.

—-

Technical corrections

- P1, L12: exits > exist. // - P1, L16: at dark spectra > for dark spectra - // P1, L 25: "so that darker bare ice is exposed" > I suggest specifying "in Summer" to be more precise. // - P2, L33: "gap of knowledge" > "knowledge gap"

—-

Will change as suggested.

—-

- P2, L39: "comparatively high resolution" > Comparatively to what? Please be more specific. Sentinel-2 and Landsat-8 could be referred to as "medium resolution sensors".

—-

We suggest changing:

"2) Compare commonly used, comparatively high resolution satellite-derived reflectance products with in situ measurements, highlighting areas in which further study is required if ongoing processes related to deglaciation are to be fully captured by satellite data."

To:

"2) Compare reflectance products derived from Landsat-8 and Sentinel-2 data with in situ measurements, highlighting areas in which further study is required if ongoing processes related to deglaciation are to be fully captured by satellite data."

The comparative statement was meant mainly in reference to the resolution of MODIS, but this is probably better expressed elsewhere in the text in a more specific manner, which we will do in the revised manuscript.

[Figure]

—-

- P2, L59: "in relatively recent times" > Please be more specific.

—-

Propose to replace with "...throughout the last decade".

—-

- P3, L86: "different kinds of remote sensing" > this phrasing is a little vague, could you clarify?

—-

Suggest changing:

"....albedo products derived from different kinds of remote sensing data..."

To:

"...albedo products derived from airborne imaging spectroscopy and Landsat and Sentinel data..."

—-

- P4, L122: "specdal" > "spectral"

—-

It should be "SpecDal" and refers to the python package we used to process the data. We will rephrase the sentence to make this clearer and cite the documentation. https://specdal.readthedocs.io/en/latest/index.html

—-

- Figure 9: please specify the wavelength of band 3.

—-

Will specify the wavelength in the figure caption.

—-

- Table 2: is lacking the first column header

—-

Will add missing column header.

—-

- Table 1, 2 and 3: I am guessing that the authors will format the tables correctly in the next iteration? They are currently unpleasant to read.

—-

Yes, we will properly format the tables.

References Reviewer

Fausto, R.S. and van As, D., (2019). Programme for monitoring of the Greenland ice sheet (PROMICE): Automatic weather station data. Version: v03, Dataset published via Geological Survey of Denmark and Greenland. DOI: https://doi.org/10.22008/promice/data/aws

Kimes, D. S., J. A. Kirchner, and W. Wayne Newcomb. "Spectral radiance errors in remote sensing ground studies due to nearby objects." Applied optics 22.1 (1983): 8-10. Malinka, Aleksey, et al. "Reflective properties of white sea ice and snow." Cryosphere 10.6 (2016).

Schaepman-Strub, G., et al. "About the importance of the definition of reflectance quantities-results of case studies." Proceedings of the XXth ISPRS Congress. 2004.

Schaepman-Strub, Gabriela, et al. "Reflectance quantities in optical remote sensingâAËŸTDefinitions and case studies." Remote sensing of environment 103.1 (2006): 27-42.

Wright, Patrick, et al. "Comparing MODIS daily snow albedo to spectral albedo field measurements in Central Greenland." Remote Sensing of Environment 140 (2014): 118-129.

Author references:

Naegeli, K., Damm, A., Huss, M., Wulf, H., Schaepman, M., & Hoelzle, M.: Cross-Comparison of albedo products for glacier surfaces derived from airborne and satellite (Sentinel-2 and Landsat 8) optical data, Remote Sensing, 9(2), 110, 2017. Naegeli, K., Huss, M., & Hoelzle, M.: Change detection of bare-ice albedo in the Swiss Alps, The Cryosphere, 13(1), 397- 412, 2019.

Di Mauro, B., Baccolo, G., Garzonio, R., Giardino, C., Massabò, D., Piazzalunga, A., Rossini, M., and Colombo, R.: Impact of impurities and cryoconite on the optical properties of the Morteratsch Glacier (Swiss Alps), The Cryosphere, 11(6), 2393, 2017.

---

## Author Response (AR1)

Author responses are below the respective reviewer/editor comments in green text. A marked-up version of the manuscript with tracked changes is attached after the responses to the comments. We have significantly changed the manuscript (rewritten and restructured the discussion, expansion of methods section, changes to figures, …) and accordingly the manuscript with the marked-up changes is rather unpleasant to read. We hope our responses to the comments are comprehensive in giving an overview of the specific changes made and suggest looking at these in combination with the non-marked up version of revised manuscript.

**Editor comments:**

I also have a couple of additional comments on the submitted MS:
- Are the 246 spectra collected 246 averages of multiple spectra ("stacked")? What is typical in ground-based albedo studies?
- A better description of the measurement protocol (identified by reviewer #2) is essential, along with context, i.e., comparison against accepted best practices for such measurements.

The spectra we present in this study are not stacked. Most other studies that follow a similar measurement protocol as ours and measure reflectance of ice/snow with a portable spectroradiometer do not use stacked spectra as such, though spectra might be averaged after grouping by surface type (e.g. Naegeli et al., 2015, 2017; Malinka et al., 2016; Hendriksa et al, 2003). In contrast, Di Mauro et al. (2017) use stacked spectra, averaging over 15 scans each time. They do not comment on how this affects their results. Our test runs in the field indicated that the measured spectra are very consistent between "shots" provided the position of the instrument is not changed so we chose not to average over multiple spectra for each point. We cannot comment in detail on typical procedures in ground-based albedo studies that deal with very different kinds of surfaces (eg vegetation) but our understanding is that it varies depending on the specific questions that are being investigated. We have expanded the description of the measurement protocol and cite previous studies where a similar approach is used.

**Reviewer 1**

Reviewer comment:

This study by Hartl et al (2020) compares a detailed field survey of albedo on Jamtalferner with synchronous remote sensing derived albedo from Sentinel and Landsat images. The methods for both approaches to albedo determination are well explained. The comparison of the field albedo and remote sensing derived albedo is the key output of this paper and is well illustrated in Figures 7-9. The study provides a richer data set for understanding how Landsat or Sentinel images could be used and is simply interesting. The primary comments below are seeking more context: 1) On the value of detailed spatial and temporal albedo observations. 2) For connections with energy balances. I am not suggesting additional data or figures be presented, but instead additional reference to other work and how the data here fits with these.
* * *
Author response:

Thank you for your comments and suggestions. We have significantly expanded the discussion to include more depth on points 1 and 2, among other considerations.

We address the specific comments below. We have attempted to follow all suggestions within the constraints of the data available to us.
* * *
9: The first sentence reverses the cause and effect. "As Alpine glaciers recede, they are quickly becoming snow free in summer and, accordingly, spatial and temporal variations in ice albedo increasingly affect the melt regime. "

Instead I suggest, "As alpine glacier become snow free in summer, recession occurs, and further spatial and temporal variations in ice albedo increasingly accentuate the melt regime."
* * *
Rephrased to: "As Alpine glaciers become snow free in summer, further spatial and temporal variations in ice albedo increasingly accentuate the melt regime and recession occurs."
* * *
16: Finishing the sentence with fluid is confusing since that could be a surface type, "Spectra can roughly be grouped into dry ice, wet ice, and dirt/rocks, although transitions between types are fluid." Maybe finish with, "although gradations between these groups occur". Replace "fluid" with gradations throughout.
* * *
Changed as suggested and checked manuscript for use of "fluid".
* * *
24: Explain that firn cover is lost when persistent loss of snow cover in the accumulation zone exposes the firn (Fischer, 2011).
* * *
Changes sentence to: "Glaciers in the Eastern Alps are losing mass rapidly, and due to persistent loss of snow cover exposing the underlying firn (Fischer, 2011), many have lost much of their firn cover."
* * *
59: ". . .relatively recent times", be more specific.
* * *
Replaced with "...throughout approximately the last decade".
* * *
74: Azzoni et al. (2016) also found a significant impact from rain water.
* * *
Changed "as well as significant effects of melt water" to: "as well as significant effects of melt and rain water"
* * *
76: What is the basis for Brun et al (2015) stating importance of remote sensing in albedo assessment?
* * *
Brun et al (2015) point out that satellite products are critical for glaciological studies in data sparse regions such as the Himalayas, where their study sites are, as in situ data are often not available and glaciers may not be easily accessible. In their study they reconstruct annual mass balance from MODIS albedo data for two glaciers, validating this with in situ data. They suggest that this method can be applied to certain other glaciers in the HKH region from which no in situ mass balance or albedo data is available. This highlights that 1) remote sensing is often the only way of getting albedo data and 2) other important glaciological work can be carried out using remote sensing derived albedo data.

Changed text as follows to makes this clearer:

"Brun et al. (2015) highlight the importance of remote sensing data for monitoring of glacier albedo changes in remote regions..."

To

"Brun et al. (2015) highlight the importance of remote sensing data for monitoring of glacier albedo changes in remote regions where data collection on the ground is impossible or impractical..."
* * *
77: Resolution of Naegeli et al. (2015) aerial albedo observations?
* * *
From Naegeli et al. 2015: "The flying altitude of 4000 m above ground level (a.g.l.) in combination with an instantaneous field of view (FOV) of 0.0025° resulted in a surface projected pixel resolution of ~ 2 m."

Changed sentence to include resolution:

"Naegeli et al. (2015) use in situ spectrometer and airborne image spectroscopy data with a pixel resolution of approximately 2m to classify glacier surface types"
* * *
96: Is it worth observing that for degree day modelling changing albedo with time would alter parameters in the model.
* * *
Added the following sentence to the paragraph: "In addition, delineating temporal variability of reflectance properties is relevant to degree day modelling, as a changing albedo would alter parameters in the model."
* * *
109: Given the illustrations in Figure 2 leverage these with terminus retreat from 1990- 2017 and for the accumulation zone what is the mean AAR during this same period 1990-1999, 2000-2009 and 2010-2017?
* * *
The requested AAR values are as follows:

1990/91-99/00: 0.35

2000/01-09/10: 0.18

2010/11-17/18: 0.12

Jamtalferner has experienced a rapid loss of firn and AAR was 0/the glacier was essentially snow free in the hydrological seasons 2002/03, 2014/15, and 2016/17. AAR values are contained in the data sets downloadable at: https://doi.pangaea.de/10.1594/PANGAEA.818772

We removed this figure in the revised manuscript based on suggestions by Reviewer 2. We added the mean AAR values for the 1990/91-99/00 and 2010/11-17/18 periods to the text.
* * *
117: "Along each profile line spectra are gathered at equal intervals, with 14 profile lines containing 11 spectra spaced at 2(?)m and 2 profiles containing 40 spectra gathered at a higher resolution of 0.5(?) m." // 132: "Google Earth Engine" // 161: "gradational" instead of "fluid"

Changed as suggested.
* * *
196: Profile 8 seems to have the least agreement in Figure 9 between field are remote sensing data, why?
* * *
We have attached photos of the profile at the end of this document to provide visual context. Profile 8 crosses a section of ice where the contrast between dark and bright areas is comparatively strong. The profile is roughly at a right angle to the flow direction and there are "stripes" of meltwater channels and/or dirt that cross the profile. The profile has a comparable number of individual spectra with reflectance values above and below the profile mean, i.e. it is not a dark profile with a few bright outliers (compare e.g. to P6 in Fig 8) or vice versa (e.g. P3), but alternates along the profile line. Agreement with the remote sensing data is decent for the darker spectra in P8 but the bright values are not captured.

While we cannot rule out that the lack of agreement between the field and remote sensing data is due to an unusually unfortunate/unrepresentative positioning of the field measurement points in the satellite pixels, this may be an instance where the diurnal melt cycle and the associated presence/absence of water on the surface exacerbates the contrast between the dark and bright sections of the profile. In the bright sections, the porous weathering crust and cryconite hole structures appear to be drained of water, while the depressions of the melt channels are noticeably wet. Cook et al. 2015 (https://doi.org/10.1002/hyp.10602) indicate the occurrence of "sudden drainage events" in the weathering crust on a day-to-day time scale and a diurnal cycle of the hydrology of the weathering crust driven by meteorological conditions (radiation, turbulent fluxes). The time of day of a satellite overpass would determine which stage of this cycle the satellite "sees" and consequently the satellite data would not capture this variability. A more definitive explanation would require further study and dedicated field experiments designed specifically to explore this aspect of reflectance variability – we hope to do this in the future.

We have added a version of the above commentary to the discussion (last paragraph of section 4.2).
* * *
204: Figure 8 has excellent potential for the direct spatial correlation of the Sentinel albedo to the point measurements. I think showing all the profiles prevents being able to visualize the relationship. I suggest focusing on a few of the same profiles that were a focus of Figure 5 and provide a range of conditions ie. P 3, 5, 8, and 11. Anzoni et al (2016) noted a future goal of generating an albedo map. Is that feasible for the area of the glacier shown in Figure 1?
* * *
Changed the figure so that only profiles 3, 5, 8, and 11 are shown. We rescaled the circles in order to give a visual representation of the horizontal uncertainty of the GPS coordinates.

Based on high resolution, close range digital images of the ice surface at Forni glacier, Anzoni et al (2016) develop a relationship between the area ratio of ice covered by fine debris and clean ice (d) such that albedo can be derived for a given area if d is known. To apply this method to the area of the glacier in Figure 1 would require an estimate of d, for which we would need close range imagery of the ice surface for the entire area. We could perhaps apply the method of Anzoni et al. to the photos we took of the ice surface at our sampling sites, but this would still result in albedo values only at our sampling sites without addressing the question of how representative these locations are for the rest of the glacier area and how albedo might be interpolated between them. We hope that our work may eventually contribute to methods for producing high resolution albedo maps, but do not think making such a map is feasible with our current dataset and the method described by Anzoni et al. Naegeli et al. 2017 produce an albedo map based on a classification of different surface types in remote sensing imagery. This would be probably be the approach of choice for Jamtalferner given the currently available data for the site.
* * *
210: This is a key observation. What have other studies found in terms of the over/under-estimate transition?
* * *
We have not been able to find many other studies with explicit information on this issue specifically over glacier ice surfaces. Hendricks et al. (2004) state for measurements at Hintereisferner: "Except for ice, the glacier reflectances derived from the satellite image are large underestimations when comparing them to the spectrometer measurements. A maximum underestimation of 139 % was found for firn in band 4. New snow, with the highest reflectance of 86 % is predicted most accurately within a confidence interval of 15 - 18 %. The reflectance of ice seems to be highly variable with both under -and overestimations of up to 76 % and 31 % respectively." This refers to Landsat ETM+ imagery acquired about 2 weeks before the corresponding field measurements. We have cited this in the revised manuscript with additional discussion of possible explanations for the location of the over/under-estimate transition. Further measurements specifically investigating this issue are needed to truly explain this effect.
* * *
226: The variation in energy balance as albedo/debris cover changes spatially and temporally was a focus of Nicholson and Benn (2006) provided a nice overview of this from Ghiacciaio del Belvedere. They observed for debris cover areas the dominant energy contribution varied from sensible heat to shortwave radiation due to decreased albedo and higher surface temperatures. They further found that for dry debris cover, sensible heat flux became negative as debris cover thickened, because of higher surface temperatures and that longwave radiation became negative even for thin debris cover.
* * *
We have added the following note in the discussion to reflect the findings on Nicholson and Benn: "Nicholson and Benn (2006) indicate that the surface albedo of ice with scattered debris can be simulated in a modelling approach be linearly varying between clean ice albedo values and values for debris, but this does not necessarily account for other types of surfaces and even the clean ice albedo can vary considerably, especially if liquid water is present."
* * *
231: How significant is the time of day variation in albedo? How consistent would this variation be from day to day? Moller and Moller (2017) provide one measure of this in an examination of spatiotemporal variations of albedo across Svalbard glaciers, recognizing this is a larger scale model albedo product. Nicholson and Benn (2012) examining Ngozumpa Glacier identify surface albedo variation across an area of varied debris cover, as well as the changing diffusivity through the melt season. The surface temperature variation of this glacier in the Himalaya would be much different than in the Alps, yet the continuous record compiled does provide context to the degree of variation and the potential importance of ongoing point measurments. They observe the importance of distinguishing wet vs dry surfaces. Azzoni et al (2016) note the increased albedo due meltwater presence during the middle of the day to albedo, while rain led to increased albedo for several days.
* * *
We have added the following paragraph to the discussion to address these points:

"Cook et al. (2016) indicate the occurrence of "sudden drainage events" in the weathering crust on a day-to-day time scale and a diurnal cycle of the hydrology of the weathering crust driven by meteorological conditions (radiation, turbulent fluxes). The time of day of a satellite overpass would determine which stage of this cycle the satellite sees and consequently the satellite data would not capture this variability. In order to assess how much time of day of the overpass could systematically affect the representativeness of the satellite date for actual ground reflectance, it needs to be determined how significant and how consistent the diurnal cycle is. To do this, the driving processes must be identified, keeping in mind that these may be different for different types of glaciers and that different causes of short-term albedo change can overlap. E.g.: Azzoni et al. (2016) point out that meltwater increases albedo around midday in a daily cycle, while rain causes increased albedo for more than one day. A seasonal cycle of albedo has been demonstrated in previous observational studies and modelling efforts of broadband albedo, which also highlight the importance of continuous measurements (e.g. Hoinkes and Wendler, 1968; Nicholson and Benn, 2012; Möller and Möller, 2017)."

In order to quantitatively answer the questions posed at the beginning of this comment, we very much hope to expand our data collection at Jamtalferner and install instrumentation that would allow continuous point measurements.
* * *
249: Similarly, the question of how well the albedo variations need to be resolved to model or understand surface processes need to be acknowledged/discussed. One reason a relatively sparse ablation stake network can represent ablation during a melt season is that despite significant surface changes the spatial distribution of energy balance over time tends to balance. Your Figure 5 illustrates this that though albedo varies considerably along the Profile 3 and 11, and the profiles have been exposed ablation ice for some period, the ice surface is relatively even. Energy balance distribution across an ice surface in a small area responds to the variations in surface level, albedo and debris cover.
* * *
"Similarly, the question of how well the albedo variations need to be resolved to model or understand surface processes need to be acknowledged/discussed." - This is a valid and interesting point of discussion. We suggest that the answer to this question depends on the processes one is trying to understand and the scale at which they occur. In the context of glacier wide ablation monitoring via the direct glaciological method, resolving sub daily and sub meter variations is perhaps not exactly a very pressing need, but, in our opinion, still interesting. The area of the glacier shown in Figure 1 contains 9 ablation stakes, which we maintain as part of our mass balance monitoring program at Jamtalferner. We observe significant differences in the amount of melt that occurs at these stakes. Aspect, shading, and slope angle of course play a strong role in this, as does locally increasing debris cover. We hypothesize that darkening due to water on the glacier surface is more of a factor at some of our stakes than at others, depending e.g. on their position in relation to seasonally shifting meltwater channels. We would like to eventually achieve a clearer separation of the influence of these factors (especially the influence of water), their relative magnitudes, and possible changes over time. We think that small scale reflectance monitoring can contribute valuable insights in this context.

Additionally, data with high spatial and temporal resolution seems essential to improve understanding of micro-hydrological processes in the weathering crust and how these may affect a possible larger scale darkening of increasingly snow free glaciers, e.g. by favoring or impeding the growth of ice algae, or the collection/washing out of cryoconite.

We have added the following section to the discussion:

**"4.3. Relevance of small-scale variability, way forward**

The reflectance properties of ice are a central part of mass and energy balance modelling, usually in the form of a glacier wide broad band albedo, or using one value for ice in the ablation zone and one for snow covered areas. Resolving local albedo variations at a very small, sub-pixel scale is not required for regional or global studies, provided the albedo parametrization captures the conditions on the ground adequately for the region of interest. In their important 2015 study, Naegeli et al. find that Sentinel-2 and Landsat-8 reflectance data are within the suggested accuracy requirements for global climate modelling (±0.05, Henderson-Sellers and Wilson, 1983) over their study site, Glacier de la Plaine Morte in Switzerland. In the same study, they report a 10% difference in modelled mass balance when a spatially distributed albedo is used to force the model as opposed to a single, glacier wide albedo. Significantly larger differences occur in parts of the glacier where water is present on the surface or the ice surface contains a lot of light-absorbing impurities. While the glacier wide impact of a spatially distributed albedo on model results may be relatively small, this highlights that resolving local variability of reflectance properties and its causes is important for accurately predicting the future evolution of individual glaciers, especially in cases where the firn covered area is gone or greatly reduced and rapid melt is occurring. Only once the problem of different scales comparing point and spatially averaged data is solved, the relationship between albedo variability and mass balance point and averaged data can be tackled to calculate the effects on mass balance at glacier-wide or regional scale.

Aside from directly mass and energy balance related applications, reflectance data with high spatial and temporal resolution is essential to improve understanding of micro-hydrological processes in the weathering crust and how these may affect a possible larger scale darkening of increasingly snow free glaciers, e.g. by favoring or impeding the growth of ice algae, or the collection/washing out of cryoconite or other impurities. High resolution time series of spectral reflectance at representative locations in the ablation zone are needed to assess how changes in wetness and temperature, surface texture (cryoconite formation, roughness changes during the season), biotic productivity, deposition of sediment by melt water and rain affect albedo on a small spatial scale, throughout the day and over the course of the ablation season. Establishing measurement efforts aimed at generating such time series on glaciers with existing mass balance monitoring networks would be highly desirable."
* * *
260: A significant source of uncertainty for what?
* * *
Surface reflectance and parameters that might be derived from it are key variables in glaciological modelling and uncertainty therein accordingly contributes to overall model uncertainty. This has implications for applications such as modelling runoff and catchment hydrology.

We have rephrased this to read "...source of uncertainty in modelling applications..."
* * *
271: Need a reference from a different region to emphasize this point.
* * *
We have removed the sentence this refers to as a part of the restructuring of the discussion and conclusion sections.
* * *
280: Did you sample spectra at any location over a period of time? If so, this helps relate the logistical challenge of temporal albedo monitoring.
* * ** * *
Profile 8, looking east:

[Figure]

Profile 8, looking west:

[Figure]

**Reviewer 2**

Reviewer comment:

In this paper, the authors present a comparison between spectral reflectance measurements of bare ice carried out in the ablation zone of the Jamtalferner glacier, Austria with concurrent Sentinel-2 and Landsat-8 acquisitions. In a first step, the spatial variability of the manually acquired surface albedo across the ablation zone of the glacier is presented, highlighting large differences in reflective properties from dry clean ice to surfaces covered in mineral and organic debris. Secondly, the paper focusses on comparing the field measurements with atmospherically-corrected satellite reflectance products to investigate whether physical processes related to deglaciation are fully captured by optical Earth Observation sensors. Results show that the differences observed between the ground-based and satellite measurements are not uniform depending on the wavelength, the sensor or surface type. The authors conclude by suggesting that further in-situ monitoring efforts are needed to be able to use satellite-derived reflectance for glacier change monitoring.

General assessment

The comparison of in-situ surface reflectance measurements with satellite-derived products is of great interest for anyone involved in space-borne observations of glaciers and more generally glacier surface processes monitoring, and in that sense, the work here is timely and most welcome. I particularly commend the use of openly accessible world-wide available satellite data rather than higher-resolution commercial data, making the applications available to a wider audience. The article is overall well written, apart from a couple of minor approximations (see detailed comments). However, the manuscript presents two major shortcomings that leave the reader missing significant information (see General comments paragraph below).

In summary, this article would have merit for publication in The Cryosphere if the major points referred to below are addressed. Currently, the Methods and Discussion sections are insufficient.
* * *
Author response:

We thank the reviewer for their time and the detailed and constructive commentary. The points of criticism are valid and we will address them in a revised version of the manuscript, following the suggestions by both reviewers.

To remedy the main shortcomings of the Methods and Discussion sections as specified in this review, we have:

1) expanded the Methods section, particularly the description of the measurement protocol for the in situ data collection.

2) restructured and significantly expanded the discussion section to address the specific issues pointed out by the reviewer in the comments below.

We address further comments individually below.
* * *
General comments

The first deficiency mentioned in the paragraph above concerns the presentation of the Methods. The ground measurements of spectral reflectance presented in Section 2.2 (7 lines) are largely insufficient for a piece of work dedicated to comparing ground measurements to satellite products. Indeed, the section barely skims over the way measurements were collected and crucial information is lacking to clearly understand the comparisons made.
* * *
See response to specific comments below.
* * *
1. When were the measurements collected? No date or time of measurements is provided in the section describing ground measurements. The reader has to wait until Section 2.3 to understand that the measurements were acquired on 4th September 2019. Over what time period (start and end of acquisitions) was the data acquired? This is of significant importance for the comparison of the data, e.g. did the surface have time to change between the satellite overpass and the ground measurements?
* * *
Ground measurements were taken on 4$^{th}$ September 2019, between approximately 10 am and 3 pm local time. The Sentinel overpass occurred at 10:20 GMT on Sept. 4. The Landsat overpass occurred at 10:10 GMT on Sept. 3. We have specified this in the revised manuscript and begin the section describing the ground measurements by stating the date and time period of the data collection. The glacier surface is constantly changing to some extent, but weather conditions on Sept. 3 and 4. were very favorable and there was no change introduced by factors such as precipitation of deposition of impurities through wind during the time period between the acquisitions. We have added information on the weather situation in the Methods section and added further comments on this in the Discussion.
* * *
2. There is no description of the environmental conditions during the acquisition, e.g cloud cover. Even a small amount of cloud cover, such as the presence of rapidly changing cirrus can introduce uncertainties of several percent in the measured reflectance.
* * *
The study site is free of cloud cover in both satellite images and the weather was sunny and dry on both days. Attached is a plot of incoming solar radiation from a weather station a short distance below the glacier showing the cloud free conditions of Sept. 3 and 4. We have included a description of the weather conditions in the revised methods section.

Figure: Incoming solar radiation at Jamtalhütte automatic weather station from September 1 2019 to September 5 2019. Data provided by the hydrology office of the state government of Tyrol, who operate this station.

[Figure]
* * *
3. The method for measuring the distance between the points on the profile is indicated, but how were the measurements geo-located in the field? Were there any GPS points acquired (especially as the authors refer to "GPS profile" in figure 1), with what uncertainty? The uncertainty in the positioning of the ground spectra may impact your point-to-pixel comparisons (to be addressed in the Discussion also).
* * *
GPS points were taken at the start and end point of each profile line, using a standard handheld GPS device. The horizontal uncertainty is < 3m. We have specified this in more detail in the expanded description of the in situ measurements and have added an approximation of the uncertainty in the point-to-pixel comparisons due to the GPS uncertainty in the results, which we comment on further in the discussion.
* * *
4. The measurement protocol is not described sufficiently, leaving the reader with a number of interrogations: how were the measurements carried out: was the ASD fibre optic handheld or placed on a device to reduce operator interference (Fig 3 in Wright et al. 2014, Kimes et al. 1983)? Did the authors use an optical lens on the fibre optic (if so, what field-of-view)? What height was the collector from the surface / spectral panel when performing the measurements? A description of how the measurements were performed is desired, or at the least, if the authors were following an existing protocol, a reference to the article is expected.
* * *
We have expanded the description of the measurements (Section 2.2). The fibre optic was handheld and used without an optical lens, at a distance of 35cm above the ground. This results in a circular field of view with a diameter of approx. 16cm. Our usage of the ASD device is similar to that of Naegeli et al. (2015, 2017) and Di Mauro et al. (2017), who carried out comparable measurements on glacier surfaces. We have cited these publications in this section.
* * *
5. The description of the processing of the raw ASD is missing. There are numerous steps to be carried out during the processing of data, including the application of instrument or spectral calibration files. In the current state, the description of the processing is too vague.
* * *
We used a feature of our instrument that saves the white reference measurement to the RAM of the instrument's computer. When this option is enabled, subsequent reflectance measurements are calculated with respect to the reference and the result of this calculation is saved to the output data file, such that there is no separate file for the reference and the output ASD files contain the calibrated values. We have added this information to the methods section.
* * *
6. The authors are not clear about the physical quantities measured. The title reads "Small scale variability of bare-ice albedo at Jamtalferner, Austria", and the author summarise the body of work on broadband and spectral albedo. However, in the methods, the field acquisitions are referred to as spectral reflectance and the (limited) description of the measurement protocol leads the author to believe that the authors are recording hemispherical–conical reflectance. The ground measurements are then compared to surface reflectance products derived from Sentinel-2 and Landsat-8. Particular care should be observed when describing remotely sensed quantities and I recommend that the authors verify inconsistencies throughout the paper. Very useful references in that sense are Schaepman-Strub et al., 2004, 2006 (besides an important corpus on the subject).
* * *
The reviewer's assessment here is correct. We have checked the manuscript for occurrences of these inconsistencies and define the quantities more clearly at the beginning of Section 1.2, referring to the works of Schaepman-Strub et al. Thank you for pointing these out, they are indeed very helpful.
* * *
The second shortfall mentioned in the overall remarks concerns the Discussion, that does not do justice to the paper. Indeed, in its current state, the section repeats the introduction and doesn't address the rich results obtained by the authors. The key points presented in the results are barely brushed past and the discussion on the limitations of the methods employed and possible explanations for the results obtained are missing. The paragraph starting P8, L247 would deserve (consequential) expanding in regard to the results obtained. By restructuring the

Discussion section, significant value could be brought to this otherwise valuable contribution to the observation of glacier ablation zones based on optical Remote Sensing.
* * *
We accept the reviewer's criticisms of the discussion and have rewritten and significantly expanded it. It now includes a section discussing the points relevant to the paragraph above, as well as other commentary that was previously lacking.

Specific comments

- P1, L14: in the Optical Remote Sensing community, ground reflectance is commonly referred to as Bottom-Of-Atmosphere (BOA) reflectance. I am not suggesting to replace the term, but maybe add a mention to BOA.
* * *
Changed sentence to include BOA in parentheses after 'ground reflectance'.

"...and are compared to the respective ground reflectance (Bottom-Of-Atmosphere) products"
* * *
- P1, L27: "The magnitude and [. . .] local production rates." > Although you go into further details later in the introduction, citations are missing here.
* * *
Added citations.
* * *
- P4, L106: Figure 2 and 3 seem irrelevant in the context of this paper that focusses on the comparison of ground and satellite acquisitions of reflectance and not the evolution of the surface properties over time. I suggest their removal, as they cloud the overall message. Rather, the satellite images (used in the study), of the glacier tongue with the profiles overlaid would be a nice addition to the paper.
* * *
Removed Figures 2 and 3 and added a new Figure 2 with two panels showing the satellite images.
* * *
- Section 2.3: Table 3 would benefit being completed with additional information on the Sentinel-2 and Landsat-8 acquisitions, such as acquisition time or the angular information (solar and viewing angles). A column with the corresponding ground measurement information would be a plus.
* * *
We have added additional columns as suggested to Table 3 (satellite acquisitions) and to Table 2 (in situ measurements).
* * *
- P5, L126: The acquisition time of Sentinel-2 is not specified: yet this information is important to investigate the differences between the measurements from both sensors.
* * *
We have specified this in the text and added it to Table 3 as suggested.
* * *
- P5, L139: Did the authors consider integrating the spectral measurements using the (available at least for Sentinel-2) spectral response of each band? Do the authors think that the difference with the average would be negligible or not?
* * *
We extracted the associated measured in-situ reflectances for each spectral range per band per sensor. Thus, the averages of the in-situ measurements can be directly compared with the reflectances per spectral band. For the comparison of Landsat 8 vs Sentinel-2 mean reflectances it should be noted that the BOA reflectances are used as they are provided by NASA and ESA, respectively, i.e. products are prepared with different radiative transfer models and different parameterizations of the atmospheric conditions. For a proper usage of the spectral response function, the L1C data should be processed using the same atmospheric correction approach and parameters. Although this is another very interesting topic, it is out of the scope of our study with the main objectives (i) analysing the spatial variability of the reflectance on a glacier's ablation zone and (ii) comparing the commonly used satellite L2A products with in-situ measurements.
* * *
- P6, L175: This is an interesting find. Have the authors considered the difference in viewing/solar geometries between the two acquisitions? The strong anisotropy of the ice could partly explain the differences (see the previous comment). Basic simulations of ice reflectance (using e.g. Malinka et al. 2016) could help investigate this point. To be clear, this is not expected from the authors, but a point that could be worth thinking about for future studies. Another factor that could influence the differences observed could be the different atmospheric corrections schemes used (a reference in the Discussion would be of value).
* * *
We have added comments on anisotropy and solar angles, as well as the issue of atmospheric correction schemes in the discussion. These factors likely contribute to the differences between Landsat and Sentinel, but without targeted further analysis and data collection it is not possible to quantify the contribution of each factor. We have also added a citation of the interesting Malinka et al. (2016) paper. Modelling reflectance properties is indeed beyond the scope of this study but would be very interesting in the future, if we can expand our monitoring situation at Jamtalferner as we hope. We believe that any modelling would have to be tuned carefully for the kind of ice surface one is dealing with, especially for very heterogenous surfaces like we have at our site. Malinka's work is based on in situ data measured on sea ice, which appears to be significantly more uniform in terms of texture and reflective properties. Their case of dark and wet sea ice still appears brighter than the majority of our spectra.
* * *
- P6, L183: This suggests that for surfaces with strong sub-pixel variability the resolution of the images is essential for an accurate description of the surface. The representativeness of field sampling when comparing in situ measurements to satellite images is of particular interest in the snow and ice community. Did the authors consider investigating the sensitivity to resolution by degrading the 10m bands to 30 then 60 meters?
* * *
We have added a comparison of differences between in situ and satellite data for the original pixel sizes and pixels resized to 30 and 60 m in the results section.
* * *
- P7, L200: Very interesting find, which links to the question of the representativeness of the in-situ sampling. It would be nice to see this point further discussed in the Discussion section.
* * *
We agree that representativeness of the in situ sampling is an important issue and discuss this in a newly added section 4.2.
* * *
- P7, L206: Again, this key result deserves some discussion.
* * *
This is now addressed in the expanded discussion section (section 4.2)
* * *
- P8, L222-226: the observation is repeated from the introduction.
* * *
Removed as a part of the rewrite of the discussion section.
* * *
- P8, L228: This paragraph should be placed in the context of the results of this study and is overall too vague.
* * *
Removed as a part of the rewrite of the discussion section
* * *
- P8, L234: Again, the paragraph reads like an introduction and doesn't have a place in the discussion.
* * *
Removed.
* * *
- P8, L244: Some lines of reflection in the context of the authors' study, such as discussing the anisotropy of ice in line with the differences in overpass geometries would be most welcome here.
* * *
We have added a discussion of these issues in Section 4.1
* * *
- Figure 4: is the highlighting of the maximum and minimum spectra necessary? A single emphasised black spectrum of the mean and the others in light grey could be clearer (if the authors agree).
* * *
Changed figure as suggested.
* * *
- Figure 6: in the printed manuscript, the tape measure is unreadable in the photos. Adding a small simple scale bar int the pictures would help grasp the scale of the images. This is an interesting figure showing the important variability of reflectance across the glacier.
* * *
Added scale bars to the figures.
* * *
- Figure 7: the caption is unclear and the reader has to read Section 3.2 several times to understand the figure. The term "ground measurements" for satellite images (P20, L419) is confusing. I would suggest revising the caption to clearly state what the blue and orange bars represent.
* * *
Rephrased the figure caption and the associated part of the text in order to improve clarity.
* * *
- Table 1: why are the PROMICE network measurements not referenced (Fausto and van As 2019)? They have been used for satellite calibration also.
* * *
Added references to PROMICE to the revised table and in the text.
* * *
Technical corrections

- P1, L12: exits > exist. // - P1, L16: at dark spectra > for dark spectra - // P1, L 25: "so that darker bare ice is exposed" > I suggest specifying "in Summer" to be more precise. // - P2, L33: "gap of knowledge" > "knowledge gap"
* * *
Changed as suggested.
* * *
- P2, L39: "comparatively high resolution" > Comparatively to what? Please be more specific. Sentinel-2 and Landsat-8 could be referred to as "medium resolution sensors".
* * *
The comparative statement was meant mainly in reference to the resolution of MODIS, but this was poorly phrased.

Changed:

"2) Compare commonly used, comparatively high resolution satellite-derived reflectance products with in situ measurements, highlighting areas in which further study is required if ongoing processes related to deglaciation are to be fully captured by satellite data."

To:

"2) Compare reflectance products derived from Landsat 8 and Sentinel 2 data with in situ measurements, highlighting areas in which further study is required if ongoing processes related to deglaciation are to be fully captured by satellite data."
* * *
- P2, L59: "in relatively recent times" > Please be more specific.
* * *
Replaced with "...throughout approximately the last decade".
* * *
- P3, L86: "different kinds of remote sensing" > this phrasing is a little vague, could you clarify?
* * *
Changed:

"....albedo products derived from different kinds of remote sensing data..."

To:

"...albedo products derived from airborne imaging spectroscopy (APEX) and Landsat and Sentinel data..."
* * *
- P4, L122: "specdal" > "spectral"
* * *
Thisshould be "SpecDal" and refers to a python package we used to process the data. Rephrased to make this clearer and added a citation of the documentation for the package. https://specdal.readthedocs.io/en/latest/index.html
* * *
- Figure 9: please specify the wavelength of band 3.
* * *
Added wavelength in the figure caption.
* * *
- Table 2: is lacking the first column header
* * *
Added missing column header.
* * *
- Table 1, 2 and 3: I am guessing that the authors will format the tables correctly in the next iteration? They are currently unpleasant to read.
* * *
We have reformatted the tables.

References Reviewer

Fausto, R.S. and van As, D., (2019). Programme for monitoring of the Greenland ice sheet (PROMICE): Automatic weather station data. Version: v03, Dataset published via Geological Survey of Denmark and Greenland. DOI: https://doi.org/10.22008/promice/data/aws

Kimes, D. S., J. A. Kirchner, and W. Wayne Newcomb. "Spectral radiance errors in remote sensing ground studies due to nearby objects." Applied optics 22.1 (1983): 8-10.

Malinka, Aleksey, et al. "Reflective properties of white sea ice and snow." Cryosphere 10.6 (2016).

Schaepman-Strub, G., et al. "About the importance of the definition of reflectance quantities-results of case studies." Proceedings of the XXth ISPRS Congress. 2004.

Schaepman-Strub, Gabriela, et al. "Reflectance quantities in optical remote sensingâA˘TDefinitions and case studies." Remote sensing of environment 103.1 (2006): 27-42.

Wright, Patrick, et al. "Comparing MODIS daily snow albedo to spectral albedo field measurements in Central Greenland." Remote Sensing of Environment 140 (2014): 118-129.

Author references:

[revised manuscript text omitted]

---

## Author Response (AR2)

Responses to the editor's comments are in green below each comment. A marked-up version of the revised manuscript is attached after the responses. Thank you for your time and comments!

12: The change in this sentence in response to Reviewer #1 is imprecise. "further spatial and temporal variations in ice albedo increasingly accentuate the melt regime and recession occurs" is evocative but unclear. The following is more straightforward: "more dark, bare ice is exposed, decreasing local albedo and increasing surface melting." Because a link between surface melt and recession is not directly shown in this MS, I recommend setting aside the latter process.

Changed as suggested.

23: "wish to" is unnecessary

Removed "wish to".

Table 1 is an especially great addition. Thanks for presenting this history.

158, 161: An in review MS should not be cited.

Removed the citations of the MS in review and changed to a citation of this data set where appropriate:

Fischer, Andrea; Markl, Gerhard; Kuhn, Michael (2016): Glacier mass balances and elevation zones of Jamtalferner, Silvretta, Austria, 1988/1989 to 2016/2017. *Institut für Interdisziplinäre Gebirgsforschung der Österreichischen Akademie der Wissenschaften, Innsbruck*, PANGAEA, https://doi.org/10.1594/PANGAEA.818772

189: "bare fibre optic" Is there a missing word here? Perhaps cable?

Changed to: "...a bare fibre optic cable without additional fore-optics".

Figures 3/4: I suggest merging these two figures, labeling the photos P3/11, and outlining both their reflectance spectra and the photos with distinct colors.

We have merged the figures and changed the figure numbering throughout the text accordingly. Please indicate if you were envisioning some other kind of arrangement for the figures, we will adjust as needed.

[Figure]

Figure 3: Each subplot on the left shows the spectra along a profile line. The bold black lines highlight the mean
spectral reflectance (HCRF) in each profile. Fotos of the ice surface along p3 and p11 are shown on the right for visual
context. Fotos were taken at the time of the respective measurements by A. Fischer.

Figure 8: What do the black diamonds signify? Add to legend.

The black diamonds are the outliers of the box plot, ie data points that fall outside of 1.5x the interquartile range. We
changed the symbol from a diamond to a + so that there is less similarity between the outlier symbol and the points
representing the landsat and sentinel data. We also added the following sentence to the figure caption: 
[revised manuscript text omitted]

451-462, 1984.